**EMBO** *reports*

# mTert induction in p21-positive cells counteracts capillary rarefaction and pulmonary emphysema

Larissa Lipskaia[1,2,12], Marielle Breau[3,12], Christelle Cayrou [3], Dmitri Churikov [3], Laura Braud [3], Juliette Jacquet[1], Emmanuelle Born [1], Charles Fouillade [4], Sandra Curras-Alonso[5], Serge Bauwens[6], Frederic Jourquin [3], Frederic Fiore [7], Rémy Castellano [8], Emmanuelle Josselin[8], Carlota Sánchez-Ferrer[9], Giovanna Giovinazzo[9], Christophe Lachaud [10], Eric Gilson [6], Ignacio Flores[9,11], Arturo Londono-Vallejo [5], Serge Adnot [1,2✉] & Vincent Géli [3✉]

## Abstract

**Lung diseases develop when telomeres shorten beyond a critical point. We constructed a mouse model in which the catalytic subunit of telomerase (mTert), or its catalytically inactive form (mTert^CI), is expressed from the p21^Cdkn1a locus. Expression of either TERT or TERT^CI reduces global p21 levels in the lungs of aged mice, highlighting TERT non-canonical function. However, only TERT reduces accumulation of very short telomeres, oxidative damage, endothelial cell (ECs) senescence and senile emphysema in aged mice. Single-cell analysis of the lung reveals that p21 (and hence TERT) is expressed mainly in the capillary ECs. We report that a fraction of capillary ECs marked by CD34 and endowed with proliferative capacity declines drastically with age, and this is counteracted by TERT but not TERT^CI. Consistently, only TERT counteracts decline of capillary density. Natural aging effects are confirmed using the experimental model of emphysema induced by VEGFR2 inhibition and chronic hypoxia. We conclude that catalytically active TERT prevents exhaustion of the putative CD34 + EC progenitors with age, thus protecting against capillary vessel loss and pulmonary emphysema.**

**Keywords** Telomerase; p21; Senescence; Emphysema; Capillary Density
**Subject Categories** DNA Replication, Recombination & Repair; Molecular Biology of Disease; Respiratory System

## Introduction

In humans, an increasing number of age-related diseases have been associated with abnormally short telomeres including aplastic anemia, pulmonary fibrosis, and lung emphysema (Armanios and Blackburn 2012; Alder and Armanios, 2022; Rossiello et al, 2022). For never smokers telomere erosion appears as a major risk factor of pulmonary fibrosis, while in smokers mutations that affect telomerase activity favor the development of emphysema and chronic obstructive pulmonary disease (COPD) (Alder et al, 2011; Alder et al, 2015; Armanios et al, 2007; Stanley et al, 2014). In mice with critically short or dysfunctional telomeres induction of DNA damage results in the development of pulmonary fibrosis (Povedano et al, 2015). In addition, telomerase-deficient mice treated with cigarette smoke develop pulmonary emphysema due to the release of inflammatory cytokines by senescent cells (Alder et al, 2011; Amsellem et al, 2011; Chen et al, 2015). More recently, it has been shown that cigarette smoke exposure in WT mice (C57BL/6NR) also induces airspace enlargement and alveolar remodeling (Railwah et al, 2020) and that aged telomerase-proficient mice exhibit a set of physiological and histological changes reflecting fibrosis-like pathology (Piñeiro-Hermida et al, 2020). These results inspired development of adeno-associated vectors (AAV) encoding the telomerase reverse transcriptase gene (*mTert*) to assess the therapeutic effect of telomerase ectopic expression either in old mice or in mice with experimentally shortened telomeres (Bernardes de Jesus et al, 2012; Povedano et al, 2018; Piñeiro-Hermida et al, 2020). Telomerase gene therapy had beneficial effects in delaying physiological aging and improving pulmonary function. However, the precise mechanism by which m*Tert* expression prevents lung damage or promotes lung endogenous repair remains to be elucidated.

[1]Institute for Lung Health, Justus Liebig University, Giessen, Germany. [2]INSERM U955 and Département de Physiologie, Hôpital Henri Mondor, FHU SENEC, AP-HP, 94010, Créteil, and Université Paris-Est Créteil (UPEC), Paris, France. [3]Marseille Cancer Research Centre (CRCM), U1068 INSERM, UMR7258 CNRS, UM105 Aix-Marseille University, Institut Paoli-Calmettes, Ligue Nationale Contre le Cancer (Equipe labellisée), Team Telomeres and Chromatin, Marseille, France. [4]Institut Curie, Inserm U1021, CNRS UMR 3347, University Paris-Saclay, PSL Research University, Orsay, France. [5]Institut Curie, PSL Research University, CNRS UMR3244, Sorbonne Université, Telomeres and Cancer, 75005 Paris, France. [6]Université Côte d'Azur, CNRS, Inserm, IRCAN, Faculty of Medicine, Nice, France. [7]Centre d'Immunophénomique, Aix Marseille Université, INSERM, CNRS UMR, Marseille, France. [8]Marseille Cancer Research Centre (CRCM), TrGET Preclinical Platform, Institut Paoli-Calmettes, Inserm, CNRS, Aix Marseille Université, Marseille, France. [9]Centro Nacional de Investigaciones Cardiovasculares Carlos III, 28029 Madrid, Spain. [10]Marseille Cancer Research Centre (CRCM), U1068 INSERM, UMR7258 CNRS, UM105 Aix-Marseille University, Institut Paoli-Calmettes, Team DNA Interstrand Crosslink Lesions and Blood Disorders, Marseille, France. [11]Centro de Biologia Molecular Severo Ochoa, CSIC-UAM, Cantoblanco, Madrid, Spain. [12]These authors contributed equally: Larissa Lipskaia, Marielle Breau. ✉E-mail: Serge.Adnot@innere.med.uni-giessen.de; vincent.geli@inserm.fr

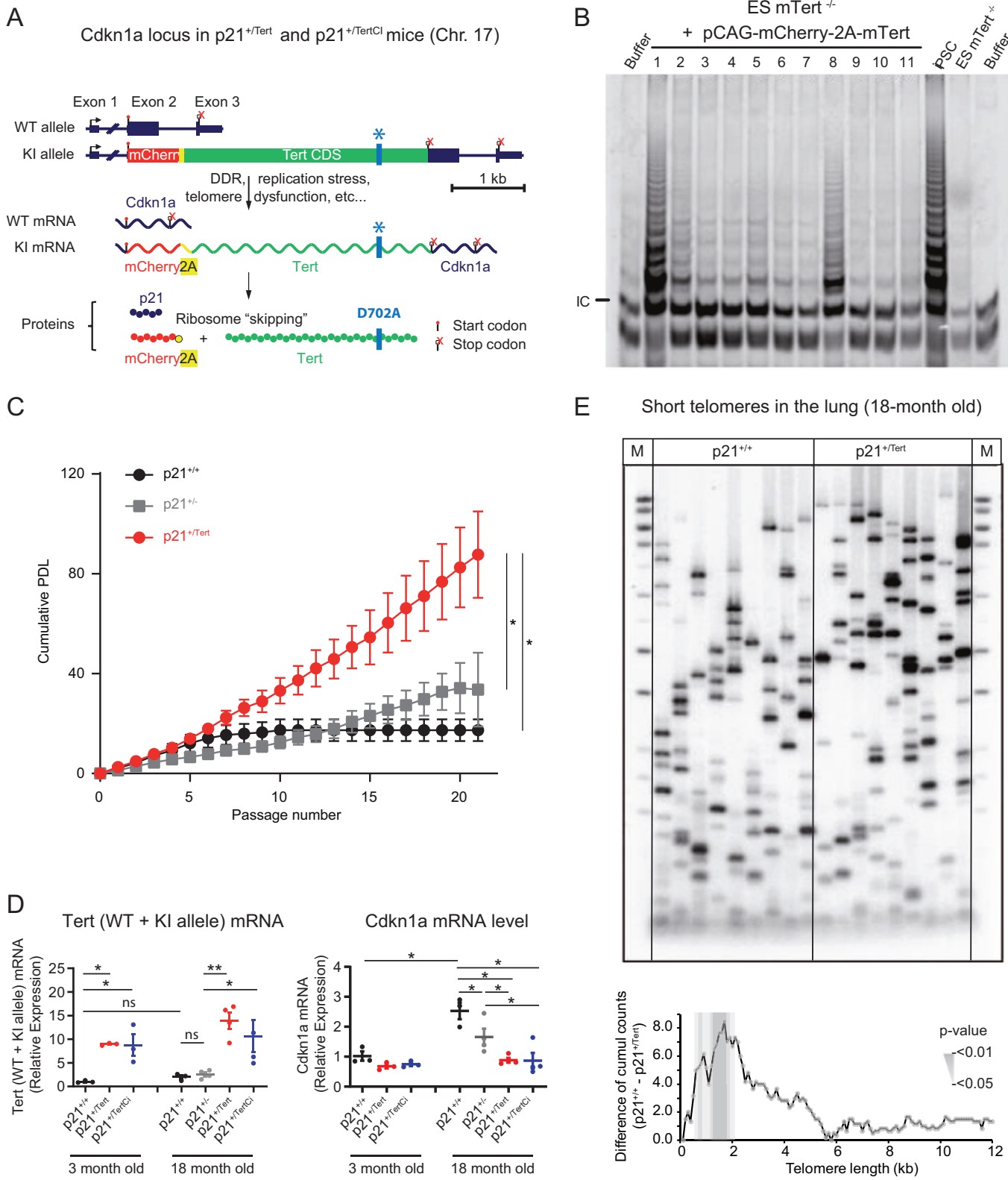

**A** Cdkn1a locus in p21⁺/Tert and p21⁺/TertCi mice (Chr. 17)

**B** ES mTert⁻/⁻ + pCAG-mCherry-2A-mTert

**C**

**D** Tert (WT + KI allele) mRNA | Cdkn1a mRNA level

**E** Short telomeres in the lung (18-month old)

**Figure 1.  Construction and validation of the p21+/Tert mouse model.**

(A) Schematic of the modified *Cdkn1a* locus, the mRNAs transcribed, and the proteins translated. The *mCherry-2A-Tert* cassette was inserted in place of the start codon of Cdkn1a (see Source data Fig. 1 for details). The gene locus is drawn to scale with intron 1 contracted. The 2A peptide sequence causes a "ribosomal skipping" that generates two independent polypeptides, mCherry and mTERT, from the same mRNA. In the *mCherry-2A-Tert^CI* cassette the codon GAT encoding D702 essential for mTERT catalytic activity has been replaced with GCA encoding A702. (B) To demonstrate that mTert expressed from the *mCherry-2A-Tert* cassette is functional, we transfected Tert-/- ES cells with a plasmid carrying *mCherry-2A-Tert* under control of the constitutively active pCAG promoter and checked telomerase activity in vitro. In vitro telomerase activity was assayed using Telomere Repeat Amplification Protocol (TRAP). The 6 bp ladder reflects telomerase activity. The arrow indicates the PCR internal control (IC). iPS and Tert-/- ES cells were used as positive and negative controls, respectively. (C) p21-promoter dependent mTert expression bypasses senescence in PA-SMCs ex vivo. Cumulative population doubling level (PDL) of PA-SMCs isolated from mice from the three mentioned genotypes. The data points are the mean values ± SD of 8 independent cultures established from individual mice. *p < 0,05 comparing p21+/Tert with p21+/+ and p21+/-, Student's t test. (D) Left panel, relative expression of *mTert* mRNA (endogenous + transgene) in the whole lung samples measured by RT-qPCR. mTert expression is shown separately for young and old mice. Right panel, relative expression of *Cdkn1a* (p21) measured by RT-qPCR in the same lung samples. Data are expressed as individual values per mice and a mean value ± SEM per group. *p < 0.05; **p < 0.01 (unpaired Student's t test). (E) Telomere shortest length assay (TeSLA) performed on lung parenchyma from p21+/+ and p21+/Tert mice. The left panel depicts 2 representative Southern blots probed for the TTAGGG repeats while the right panel shows the difference of cumulative number of short telomeres between lungs from p21+/+ and p21+/Tert mice. This difference is significant for telomeres size range between 0.4 and 2.0 kb. Southern blots for all mice are shown in Fig. EV3. Source data are available online for this figure.

In response to various stresses including telomere shortening, the p53-dependent up-regulation of p21 expression is thought to be the primary event inducing replicative senescence (Harper et al, 1995; Smogorzewska and de Lange, 2002; Herbig et al, 2004). p21 is a cyclin-dependent kinase (CDK) inhibitor that interacts with multiple CDK complexes, with a higher affinity for CDK4/2, involved in the G1/S transition (Harper et al, 1995). In normal conditions, the levels of p21 are also regulated by several additional transcription factors including E2F1, STAT3, and MYC (Abbas and Dutta, 2009). One major effect of p21 in stem cell compartments is to halt cell proliferation. Reciprocally, mice with p21 deletion present an increased number of stem cells under normal homeostatic conditions, however their number declines with age suggesting that p21 is important for their life-long maintenance (Cheng et al, 2000; Kippin et al, 2005). Since dysfunctional telomeres signal cell cycle arrest via the ATM-p53-p21 pathway (Smogorzewska and de Lange, 2002; Herbig et al, 2004), we generated a transgenic model in which the telomerase is expressed under the p21 promoter regulation. The p21+/Tert model is expected to upregulate *mTert* expression in response to telomere dysfunction, but also in response to other cues inducing p21.

Age-related decline of pulmonary function is a paradigm for telomere dysfunction-driven degenerative process characterized by p21 and p16 activation and cellular senescence (Barnes et al, 2019). Here, we show that p21 promoter-dependent expression of m*Tert* prevents emphysema in aged mice, and this coincides with maintenance of microvascular density, decreased senescence of endothelial cells, and preservation of a pool of endothelial CD34+ cells endowed with proliferative capacities. In addition, p21 promoter-dependent expression of m*Tert* also protects the lungs of young mice from endothelial cell senescence and severe emphysema induced by VEGF receptor blockade combined with hypoxia.

## Results

### Generation and validation of the p21+/Tert mouse model

To generate a mouse model that expresses m*Tert* under control of the p21^Cdkn1a promoter, we substituted the start codon of the *Cdkn1a* gene by a *mCherry-2A-Tert* cassette (Fig. 1A). The detailed construction of the targeting vector and integration of the cassette are shown in the Appendix Fig. S1. The *mCherry-2A-Tert* allele retains regulatory 5' and 3'UTRs of the endogenous *Cdkn1a* gene. Translation of the polycistronic *mCherry-2A-Tert* mRNA produces two separate mCherry-2A and mTERT polypeptides due to the ribosome skipping at the 2A sequence (Fig. 1A). The generated p21+/Tert mouse therefore produces p21 protein from one allele and mCherry and mTERT from the other. We created in addition another mouse line (p21+/Tert-CI) identically designed but with a point mutation (D702A) in the active site of *mTert* which abolishes telomere elongation by telomerase (Barnes et al, 2019; Lingner et al, 1997; Fig. 1A). The catalytic activity of mTERT expressed from the *mCherry-2A-Tert* cassette was validated by TRAP assay in the extracts from m*Tert*-/- mouse ES cells transfected with the *mCherry-2A-Tert* plasmid (Fig. 1B). The functionality of the p21+/Tert mouse model was verified by following the whole-body fluorescence emitted by the mCherry after exposure to ionizing radiation, a condition known to induce p53-dependent expression of p21 (Fig. EV1A). The induction of p21 and mCherry expression was directly demonstrated by treating p21+/Tert mice with doxorubicin followed by immunoblotting and in situ fluorescence analysis of the liver and kidney, the organs which accumulate doxorubicin (Fig. EV1B).

To demonstrate the effect of p21 promoter-dependent m*Tert* expression during replicative senescence, we examined the proliferation of cultured lung pulmonary artery smooth muscle cells (PA-SMCs) isolated from young control and transgenic mice (Fig. 1C). PA-SMCs can be readily isolated from mouse lungs and cultured. As many other mouse cell types, PA-SMCs senesce quickly in standard cell culture conditions (21% atmospheric oxygen) and cell proliferation arrest is thought to be caused by oxidative DNA damage (Parrinello et al, 2003), possibly affecting the very long mouse telomeres. As expected, PA-SMCs isolated from p21+/+ and p21+/- mice entered senescence after a few passages, with a final mean population doubling levels (PDLs) of 15.51 (+/−4.14) and 36.09 (+/−8.30) respectively, the difference between the two mice being likely due to p21 haplo-insufficiency. In contrast, the cells from p21+/Tert mice never entered senescence and proliferated at a much faster rate (Fig. 1C). The difference in cumulative PDL between p21+/Tert and both p21+/+ and p21+/- became significant at passage 7. We confirmed the expression of the *mCherry-2A-Tert* transcript by RT-qPCR (Fig. EV2A,B).

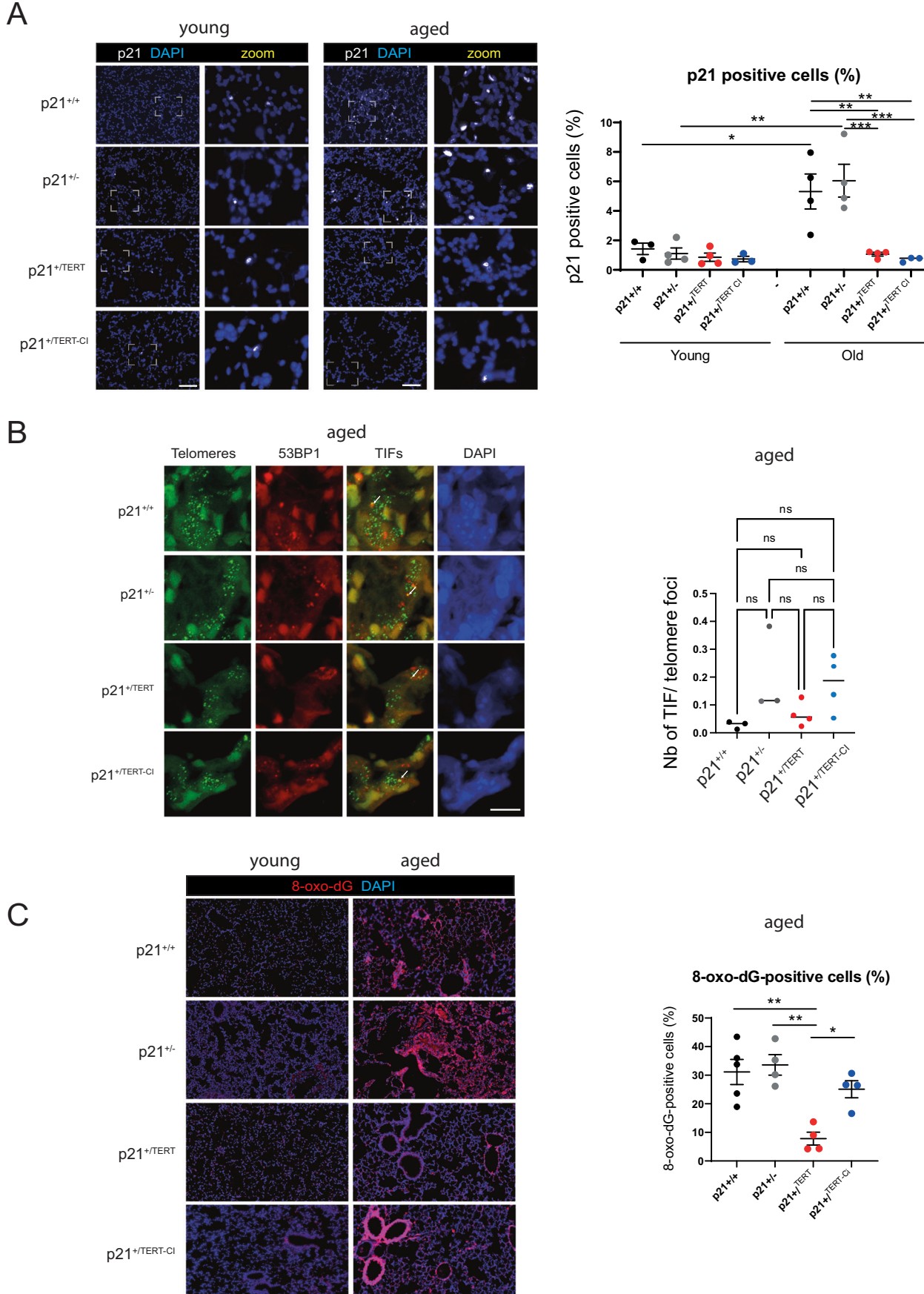

**Figure 2. p21$^{+/Tert}$ and p21$^{+/TertCI}$ mice have reduced levels of p21 in lung parenchyma but similar levels of damaged telomeres.**

(A) Left, Representative micrographs showing immunofluorescence of p21 (white) in lung cells. The zoomed areas are indicated by rectangle. Blue—DAPI nuclear staining. Scale bar—100 μm. Right, quantification of the percentage of DAPI-stained nuclei with a p21 foci. Quantification of a minimum of 10 images per mice are shown. *$p < 0.05$, **$p < 0.01$, ***$p < 0.0001$ according to two-way ANOVA test. (B) Left, Representative images of telomere FISH (green), 53BP1 immunodetection (red) and nucleus stained with DAPI (Blue) in lung section. Images represent the maximum intensity projection of the 5 μm section taken with a ×60 oil objective. White arrows indicate colocalization. Scale bar—10 μm. Right, Quantification in a lung section of the percentage of telomere colocalizing with 53BP1 done in 1 mm² corresponding to a 4 × 4 tiling image. Lung section form at least 3 mice per conditions were analyzed. Over 10,000 telomeres where quantified per mice using Nikon NIS.ai. ns (not significant), according to two-way ANOVA followed by Tukey's multiple range test. (C) Left, representative micrographs showing immunofluorescence of 8-hydroxy-2'-deoxyguanosine (8-oxo-dG) (red) in lung cells. Blue—DAPI nuclear staining. Scale bar—100 μm. Right, quantification of the percentage of 8-oxo-dG-stained nuclei. Lung sections from at least 4 mice per group were analyzed and the mean of the quantification of a minimum of 7 images per mice are shown. *$p < 0.05$, **$p < 0.01$ (one-way ANOVA test with Bonferroni correction). Source data are available online for this figure.

Importantly, we also detected a twofold higher telomerase activity in p21$^{+/Tert}$ cells at passages 2–4, before PA-SMCs cumulative PDL curve became significantly different (Fig. EV2C) indicating that upregulation of mTERT alone is sufficient to increase telomerase activity. We explain the apparent discrepancy between the high level of transgene induction and only twofold increase in telomerase activity by the fact that telomerase biogenesis and trafficking require numerous factors that can be limiting. We further measured the load of very short telomeres (VSTs) that may arise in cultured PA-SMCs causing their arrest by TeSLA (Lai et al, 2017; Fig. EV2D). While the cumulative number of VSTs increased with passages in p21$^{+/+}$ cells, it did not change in p21$^{+/Tert}$ cells (Fig. EV2E). Thus, we concluded that cultured mouse cells do accumulate VSTs coincidently with proliferation arrest, and that ectopic mTert expression can effectively counteract accumulation of VSTs coincidently with abrogation of the arrest.

We next analyzed mTert (and mTert$^{CI}$) as well as p21 expression in the lungs of 18 month-old mice by RT-qPCR (Fig. 1D). We found that mTert and mTert$^{CI}$ were expressed in p21$^{+/Tert}$ and p21$^{+/TertCI}$ mice, respectively (Fig. 1D, left panel). Interestingly, p21 transcript was reduced in the lungs of old p21$^{+/Tert}$ mice relative to age-matched WT and p21$^{+/-}$ controls (Fig. 1D, right panel). To our surprise, p21 transcript level was also reduced in the lungs of the p21$^{+/TertCI}$ suggesting that p21 expression is attenuated in a way that is independent of mTERT catalytic activity (see Fig. 2 and "Discussion").

We asked whether mTert controlled by p21 promoter would also curb the accumulation of VSTs in vivo by measuring the load of VSTs in the lung parenchyma of 18 month-old p21$^{+/+}$ and p21$^{+/Tert}$ mice (Figs. 1E and EV3A). We found that lungs from p21$^{+/Tert}$ compared to p21$^{+/+}$ mice harbored significantly less VSTs in the 0.4–2.0 kb range (Fig. 1E) suggesting that mTert expression from p21 promoter in old mice is able to partially heal critically short telomeres in vivo. We then compared in another experiment the amount of VSTs between p21$^{+/-}$ and p21$^{+/TertCI}$ old mice. As expected, catalytically inactive mTERT had no effect—no significant difference in the number of VSTs was found between these two groups of mice (Fig. EV3B). It is worth noting that mice of all 4 genotypes show no significant differences in the average length of their telomeres, which is around 20 kb (Appendix Fig. S2).

## p21$^{+/Tert}$ and p21$^{+/TertCI}$ mice have reduced levels of p21 in lung parenchyma

The RT-qPCR (Fig. 1D) revealed that p21 mRNA levels were reduced in lungs of p21$^{+/Tert}$ and p21$^{+/TertCI}$ old mice which at first glance is rather surprising. We therefore sought to analyze p21 protein levels on the lung sections of young and old mice using a

validated anti-p21 antibody. We observed that p21 labeling was very low in young mice of all 4 genotypes, but it was much more pronounced in the old p21$^{+/+}$ and p21$^{+/-}$ mice (Fig. 2A), consistent with the fact that p21 labeling increases in aged tissue. Surprisingly and counter-intuitively, p21 levels were greatly reduced in the lungs of not only the p21$^{+/Tert}$ but also in the p21$^{+/TertCI}$ old mice (Fig. 2A upper and lower panels). This reduction was not due to the haplo-insufficiency of p21 in the p21$^{+/Tert}$ and p21$^{+/TertCI}$ mice since the decrease in p21 levels in both mice is more pronounced than that in the p21$^{+/-}$ mice. We infer that a non-canonical function of mTERT negatively regulates p21 levels.

In keeping with these results, we sought to determine the level of telomeric damage in lung parenchyma of old mice of the 4 genotypes by analyzing the number of Telomere Dysfunction Induced Foci (TIF). We performed immunofluorescence staining against 53BP1 along with telomere specific FISH on lung sections from old WT, p21$^{+/-}$, p21$^{+/Tert}$, p21$^{+/TertCI}$ mice. More than 20,000 telomeres/mouse distributed in the lung parenchyma were analyzed for each of the 3 mice of the 4 genotypes. We were unable to detect a significant difference in the percentage of TIFs between the lungs of WT, p21$^{+/-}$, and p21$^{+/Tert}$ mice (Fig. 2B). Only the lungs of aged p21$^{+/TertCI}$ mice tend to have an increased percentage of TIFs. Also, we were unable to detect a difference in the global 53BP1 staining between the genotypes (Source data Fig. 2B, 53BP1 foci). Of note, although 53BP1 is a genuine marker of DNA double strand breaks, it may not be ideal for detection of damaged telomeres in the lungs, which experience a high level of oxidative stress (Zglinicki, 2002). We therefore analyzed the global level of oxidative DNA damage by measuring 8-oxo-dG levels using an anti-8-oxo-dG antibody. In old mice, 8-oxo-dG signals occurred mainly in vessels and around bronchial tubes, and to a lesser extent in the parenchyma (Fig. 2C). We found that the global level of oxidative damage was reduced in the lungs of p21$^{+/Tert}$, but not p21$^{+/TertCI}$ mice (Fig. 2C). These results demonstrate that only catalytically active mTERT can reduce the global level of oxidative damage in the lungs. Since telomere damage can trigger mitochondrial dysfunction leading to enhanced ROS activation and global oxidative stress (Passos et al, 2010), we envision that mTERT reduces both telomeric and non-telomeric 8-oxo-dG. However, the reduction in p21 levels depicted in Fig. 2A cannot be attributed solely to the reduction of oxidative damage, since it is also observed in p21$^{+/TertCI}$ mice lungs.

## p21 promoter-dependent expression of mTert protects against age-related emphysema and perivascular fibrosis

We sought to determine the impact of the conditional expression of mTert on age-related manifestations in the lung, such as

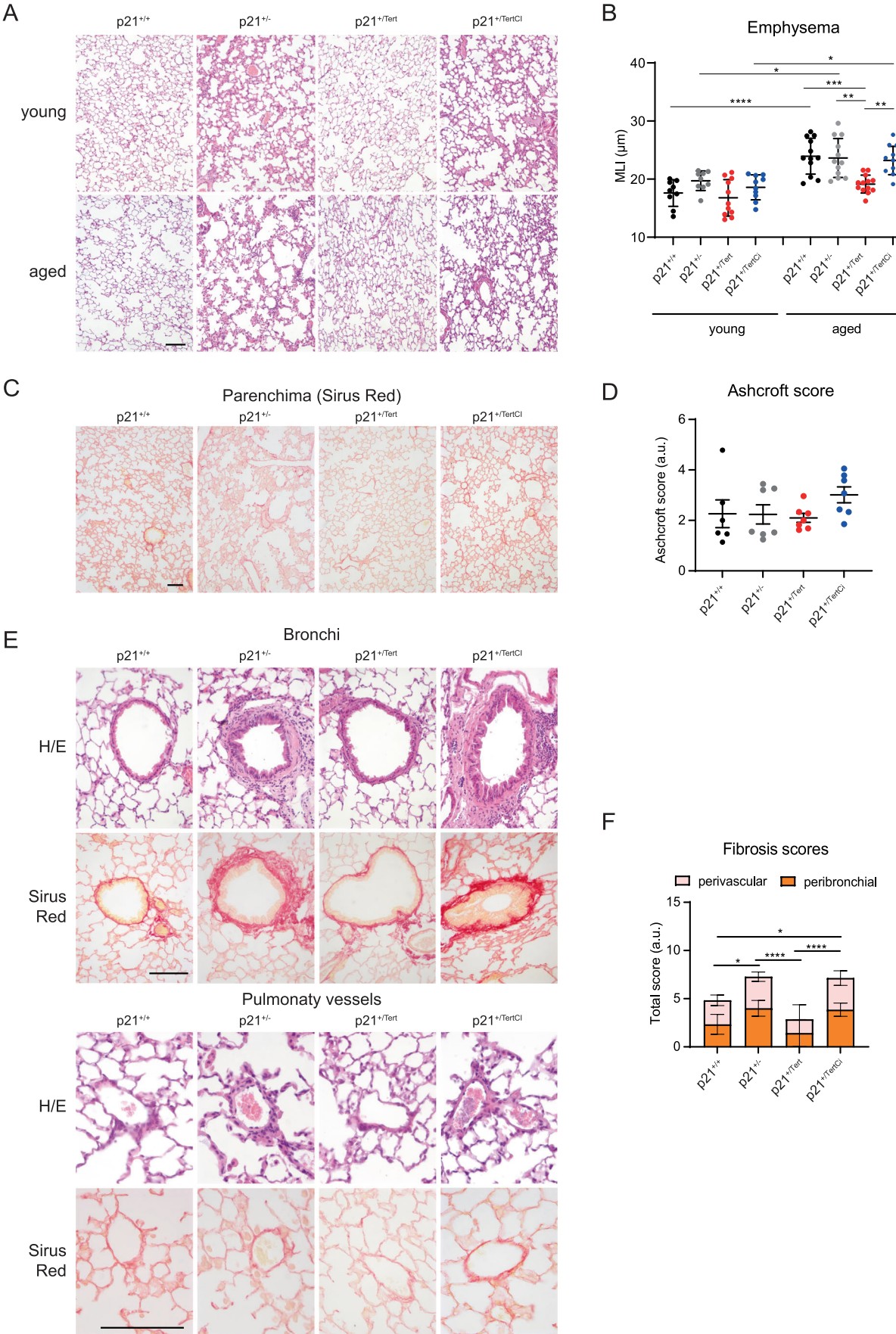

**Figure 3.   Telomerase protects against age-related emphysema and perivascular fibrosis.**

(A) Representative micrographs showing lung parenchyma of young and old mice from the four mouse models. Hematoxylin/eosin staining. (B) Scatter plot showing mean liner intercept (MLI) in young and old mice. Data are expressed as individual values per image and a mean value ± SD per group. (C) Representative micrographs showing lung parenchyma of the aged mice stained with Sirius Red to visualize collagen deposition (red). (D) Scatter-plot graph showing parenchymal fibrosis quantification according to Aschcroft score. Data are expressed as individual values per image and a mean value ± SD per group. (E) Representative micrographs showing bronchi and pulmonary vessels of aged mice stained with H/E or Sirius Red. (F) Bar-graph showing perivascular and peribronchial fibrosis scoring. Fibrosis scores were attributed on 1-5 scale: 0—absent; 1—isolated mild fibrotic changes, 2—clearly fibrotic changes; 3—substantial fibrotic changes, 4—advanced fibrotic changes, 5—confluent fibrotic masses. Lung sections from 4 to 8 mice per group were analyzed. Data are expressed as mean ± SEM. Data information: For all graphs, *$p < 0.05$, **$p < 0.01$, ***$p < 0.001$, ****$p < 0.0001$ (One way ANOVA with Bonferroni correction). For (D), no significative differences. For all images, Scale bar—50 µm. Source data are available online for this figure.

emphysema and fibrosis. C57BL/6NR mice used in this study naturally develop age-related emphysema and mild fibrosis. Emphysema is characterized by lung airspace enlargement as a consequence of a decrease of lung elasticity with advanced age (Sharma and Goodwin, 2006). Morphometry studies (Dunnill, 1962) did not reveal air space enlargement in young mice (Fig. 3A,B). In contrast, old p21$^{+/+}$, p21$^{+/-}$, and p21$^{+/TertCI}$ mice developed emphysema reflected by an increase of alveolar size (measured by the mean-linear intercept or MLI). Remarkably, old p21$^{+/Tert}$ mice did not exhibit air space enlargement (Fig. 3A,B). The fact that p21$^{+/Tert-CI}$ mice exhibit the same aging characteristics in the lung as control mice supports the idea that mTERT's effect in emphysema protection is related to its ability to elongate critically short or damaged telomeres (Matmati et al, 2020; Birch et al, 2015; Fouquerel et al, 2019).

We also searched for manifestations related to lung fibrosis in the mouse models. We did not observe any differences in fibrosis in the parenchyma among the four mouse models (Fig. 3C,D). However, Sirius Red and HE stainings revealed local increase of fibrosis around bronchi and vessels in lungs from the old p21$^{+/-}$ and p21$^{+/TertCI}$ mice compared to WT (Fig. 3E,F). In contrast, fibrosis was decreased in lungs from p21$^{+/Tert}$ mice compared to WT (Fig. 3E,F).

Collectively, these results indicate that p21-dependent expression of mTert protects mice from age-related emphysema and locally from perivascular and bronchial fibrosis.

## p21 is preferentially expressed in lung endothelial cells

To understand how *Tert* expression controlled by p21 promoter exerts its beneficial effect in the ageing lung, we determined in which lung cell types p21 is expressed. To this end, we performed single-cell RNA sequencing (scRNA-seq) on whole lungs from 18-month-old p21$^{+/+}$, p21$^{+/-}$, p21$^{+/Tert}$ and p21$^{+/TertCI}$ mice. We carried out dimensionality reduction and unsupervised cell clustering to identify distinct cell types based on shared and unique patterns of gene expression (see "Methods"). For each mouse, clustering of gene expression matrices identified cell types that were in good agreement with the published Mouse Cell Atlas (MCA) (Han et al, 2018; Fig. 4A). As shown in Fig. EV4 and Dataset EV1, similar cell types were isolated from the lung of the mice of the four genotypes (WT ($n = 1$); p21$^{+/-}$ ($n = 3$), p21$^{+/Tert}$ ($n = 5$), and p21$^{+/TertCI}$ ($n = 2$)) suggesting that Tert expression does not change drastically the lung cell population composition.

We then analyzed p21 expression in different cell types in mice of the four genotypes. Among immune cells, p21 was mainly expressed in interstitial and alveolar macrophages, monocytes, and in dendritic cells, while among non-immune lung cells, it was

mainly expressed in endothelial cells (ECs) as compared to other cell types including AT1 and AT2 cells (Fig. 4B). Unfortunately, we could not directly detect the mCherry-2A-mTert transcript because the 10× Genomics single-cell 3′ chemistry only provides sequence information on the short region preceding the polyA-tail. Since p21 (and hence mTert and mTert$^{CI}$ in p21$^{+/Tert}$ and p21$^{+/TertCI}$ mice, respectively) is preferentially expressed in pulmonary ECs we focused our single-cell analysis on this cell type. We annotated lung ECs to the specific vascular compartments based on recently published markers of lung ECs (Kalucka et al, 2020), and identified 5 separate clusters comprising two types of capillary, artery, vein, and lymphatic ECs (Fig. 4C,D). Capillary class 1, marked by *Car4*$^{high}$, was recently identified as aerocytes (aCap) involved in gas exchange and trafficking of leukocytes, and capillary class 2 as general capillary cells (gCap) that function in vasomotor tone regulation and in capillary homeostasis (Gillich et al, 2020). In the other single cell study (Niethamer et al, 2020), the *Car4*$^{high}$ capillary ECs were found to co-express a putative stem cell marker CD34, and were poised to contribute to regeneration. We uncovered that p21 is expressed at the highest level in the *Car4*$^{high}$ capillary ECs (Fig. 4E). This result is in agreement with Tabula Muris Senis (https://tabula-muris.ds.czbiohub.org; Fig. EV5). Therefore, the beneficial effects observed in ageing lung of the p21$^{+/Tert}$ mice could stem from mTert expression in the *Car4*$^{high}$ capillary ECs, possibly marked by CD34.

## mTert expression controlled by p21 promoter counteracts senescence of the lung endothelial cells

We next investigated whether mTert could suppress age-related cellular senescence in the lungs, and if so, of which cell types. For this, we analyzed the colocalization of the conventional senescence marker p16 with either the endothelial marker CD31 (Pecam-1) or the alveolar epithelial type II cell marker Muc1 in lung sections from 18-month-old mice of the 4 genotypes (Fig. 5). Single-cell RNA-seq carried out on lungs of the 4 mouse models confirmed the specificity of these markers. Importantly, we validated anti-p16 antibody by checking that p16 staining detected after an injury (leading to the induction of p16) disappeared after inducible elimination of the cells expressing p16 (see Born et al, 2023). We found that only 10% of the p16-positive cells expressed Muc1 and p21-dependent expression of mTert did not reduce the fraction of p16- positive cells marked by Muc1 (Fig. 5A,B). In contrast, we found that about 30% of the p16-positive cells were also positive for the endothelial marker CD31 (Fig. 5C,D) indicating that endothelial cells largely contributed to senescence in the lungs of the old mice. Remarkably, p21$^{+/Tert}$ lung samples accumulated significantly

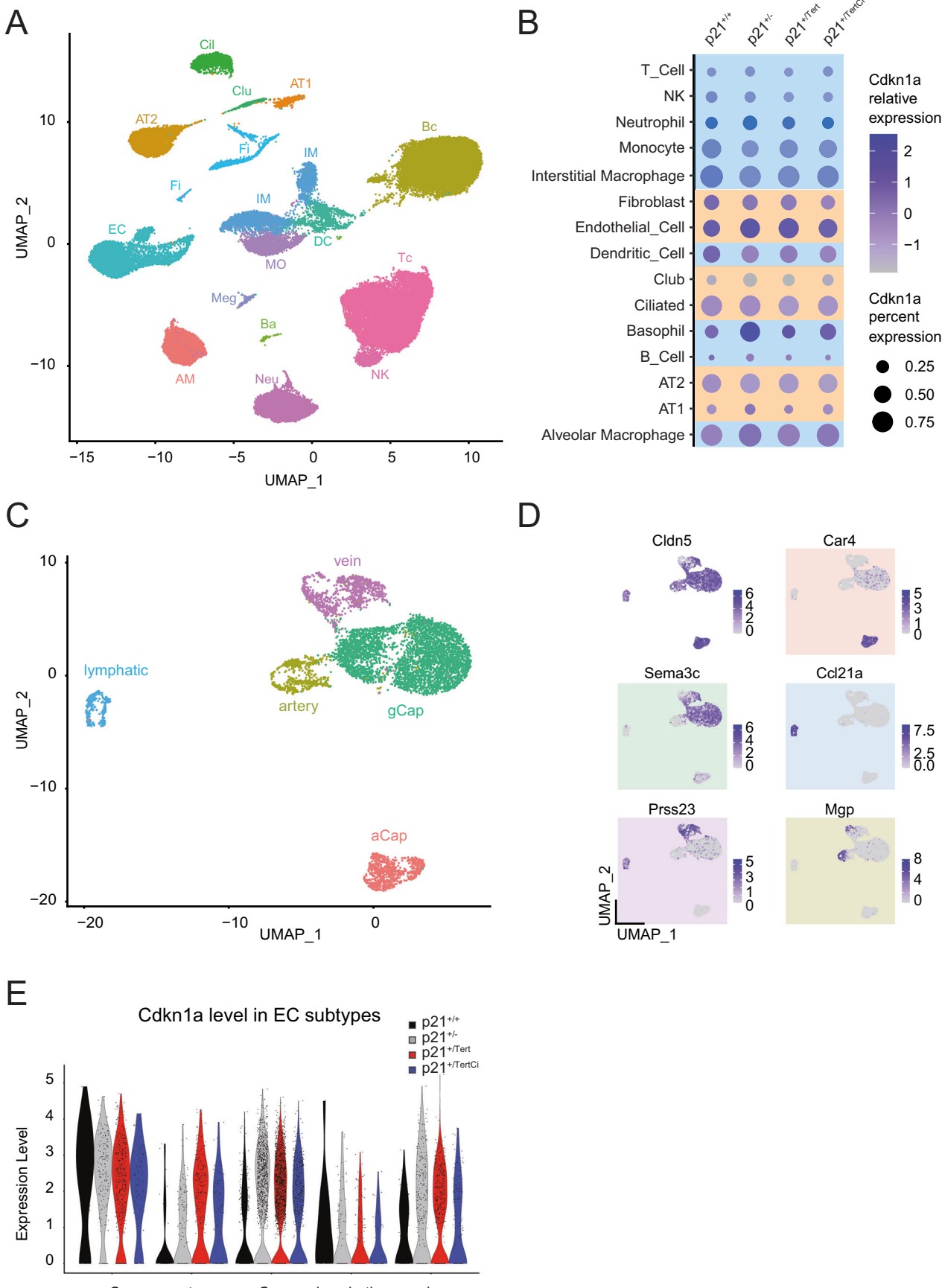

**Figure 4. p21 is preferentially expressed in lung endothelial cells.**

Lung samples from 18 month-old WT (p21$^{+/+}$), p21$^{+/-}$, p21$^{+/Tert}$ and p21$^{+/TertCI}$ mice were analyzed. For all panels WT ($n = 1$), p21$^{+/-}$ ($n = 3$), p21$^{+/Tert}$ ($n = 5$), p21$^{+/TertCI}$ ($n = 2$). (A) Unsupervised Uniform Manifold Approximation and Projection for Dimension Reduction (UMAP) clustering of lung cells. Lung cell populations were identified using Mouse Cell Atlas (MCA) annotation procedure. (B) Dot plots of *Cdkn1a* expression in the different lung cell-types. The identified cell types are shown on the y-axis. The size of the dots represents the fraction of the cells expressing *Cdkn1a*. The color intensity represents the average expression level in p21-positive cells. Immune and constitutive lung cells are color-coded in blue and orange, respectively. (C) UMAP clustering of lung endothelial cells. Cell populations were identified based on known markers for these endothelial cell subtypes. (Art) artery EC cells; (Cap_1; also called aCap) capillary 1 EC cells; (Cap_2; also called gCap) capillary 2 EC cells; (Lym) lymphatic EC cells and (Vein) vein EC cells. (D) Representative markers used to annotate lung endothelial cell classes. (E) Violin plots showing p21 expression. ECs are subdivided into groups according to cluster and genotype (WT (black); p21$^{+/-}$ (gray), p21$^{+/Tert}$ (red) and p21$^{+/TertCI}$ (blue). Source data are available online for this figure.

less p16-positive CD31 cells (Fig. 5C,D) suggesting that either mTert expression counteracts senescence of endothelial cells or senescent endothelial cells are more efficiently eliminated in these mice.

To further confirm the senescence suppression by the conditional expression of m*Tert*, we also assessed the global accumulation of senescent cells in the lungs of 18-month-old mice of the 4 genotypes by evaluating the number of lung parenchymal cells with senescence associated β-galactosidase activity (SA-β-Gal) (Fig. 5E,F). While WT (p21$^{+/+}$), p21$^{+/-}$, and p21$^{+/TertCI}$ mice displayed a significant increase in the percentage of SA-β-Gal with age, the percentage of SA-β-Gal-positive cells increased only slightly in p21$^{+/Tert}$ mice (Fig. 5E,F).

### mTert expression counteracts the age-related decline in capillary density and maintains a high number of CD34+ cells in the lungs of old mice

Microvasculature density has been shown to decrease in organs of aged mice (Grunewald et al, 2021). Because conditional induction of mTert has been reported in vivo to cause the long-term proliferation of several types of stem cells (Sarin et al, 2005; Shkreli et al, 2011; Montandon et al, 2022), we determined whether m*Tert* expressed from p21 promoter in the lungs of aged mice could affect capillary density that declines with age. We labeled lung parenchyma with CD31 to reveal the vasculature compartment in young and old mice of the 4 genotypes (Fig. 6A). Quantification of the labeling revealed a higher microvasculature density in the aged p21$^{+/Tert}$ mice compared to the 3 other genotypes (Fig. 6B). These results demonstrate that mTERT (but not mTERT$^{CI}$) counteracts the decline in capillary density in the lungs of aged mice.

The CD34 marks a subtype of capillary cells involved in healing capillaries after acute lung injury (Ding et al, 2011; Niethamer et al, 2020; Wang et al, 2022). We thus examined lung parenchyma from young and old mice for the presence of cells marked by CD34. Staining of lung sections from young and old mice for the presence of cells expressing *Cd34* revealed that young mice harbored more CD34+ cells as compared to the old mice (Fig. 7A). In young mice, the level of CD34 was higher in the lungs of p21$^{+/-}$, p21$^{+/Tert}$, and p21$^{+/TertCI}$ mice. Because p21 is involved in maintaining stem cells in quiescence (Cheng et al, 2000; Kippin et al, 2005), it is possible that even in young mice, slight differences in p21 levels in the lungs between WT and mice of the other 3 genotypes explain why young p21$^{+/-}$, p21$^{+/Tert}$, and p21$^{+/TertCI}$ mice have slightly elevated levels of CD34+ cells in the lungs. Strikingly, with age CD34+ cells were hardly detected except for the p21$^{+/Tert}$ mice in which the level of CD34$^+$ cells was similar to that in young mice (Fig. 7B). These results suggest that expansion of CD34+ cells that might be due to

reduced p21 levels is likely limited by replicative senescence in old mice, except in the p21$^{+/Tert}$ mice.

At lower magnification, we could clearly observe a population of CD34+ cells surrounding vessels in old WT and p21$^{+/Tert}$ mice and another population of CD34+ in the lung parenchyma mainly present in the old p21$^{+/Tert}$ mice (see Appendix Fig. S3). Overall, old p21$^{+/Tert}$ mice maintained this high level of CD34-expressing cells while it dropped drastically in aged mice of all other genotypes. These results raised the question of the identity of the CD34+ cells in the lungs from 18 month-old mice.

We turned to scRNA-seq of the lungs from 18-month-old p21$^{+/+}$, p21$^{+/-}$, p21$^{+/Tert}$, and p21$^{+/TertCI}$ mice. We found that *Cd34* was mainly expressed in ECs (64.9% of *Cd34*+ lung cells) and in fibroblasts (26.2% of *Cd34*+ lung cells) (Fig. 7C). We further looked at the expression of *Cd34* in the different subclasses of ECs and fibroblasts. We found that *Cd34* was expressed at highest levels in *Ebf1*+ fibroblasts and mesothelial cells (>85% are *Cd34*+ cells), two populations of fibroblasts delineated by early B-cell factor 1 and *Upk3b* expression, respectively (Sidney et al, 2014; Korsunsky et al, 2022; Fig. 7D). In ECs, *Cd34* expression was maximum in the cells defined as aCap by Gillich et al (2020) (aerocytes; >90% are *Cd34*+ cells) but also expressed in gCap and vein cells but at lower levels (between 25% to 29% are *Cd34*+ cells respectively) (Fig. 7D). However, mRNA level of *Cd34* in the different classes of lung ECs of the 4 genotypes did not reveal major differences (Fig. 7E), in apparent contradiction with the immunostaining (see "Discussion"). We thus sought to compare the distribution of *Cd34*+ cells in p21$^{+/-}$, p21$^{+/Tert}$, and p21$^{+/TertCI}$ mice (Fig. 7F). We found that in comparison with p21$^{+/-}$ and p21$^{+/TertCI}$ mice, the p21$^{+/Tert}$ mice have a higher number of aCap cells expressing *Cd34* (Fig. 7F).

### CD34+ endothelial cells show proliferative capacity in the lungs of old mice

We sought to determine whether the CD34+ cells were endowed with proliferative capacity. We stained lung cells with antibodies against CD34 and PCNA. We found that p21$^{+/Tert}$ old mice had a higher number of CD34+ cells labeled with the proliferation marker PCNA compared to the control mice while this difference was much less pronounced in young mice (Fig. 8A, left and right panels). To further assess CD34+ cell proliferation more directly, 18-month-old mice were injected with bromodeoxyuridine (BrdU) *intra-peritoneally* 24 h before lung sampling. Although the number of BrdU positive cells was low, we found a higher number of BrdU-positive cells in the lungs of p21$^{+/Tert}$ mice compared to the WT (p21$^{+/+}$), p21$^{+/-}$, and p21$^{+/TertCI}$ mice (Fig. 8B,C). Interestingly, most of the BrdU-positive cells were also labeled with CD34 (Fig. 8C). These results show that p21 promoter-dependent expression of mTert endows CD34+ cells with proliferative capacity.

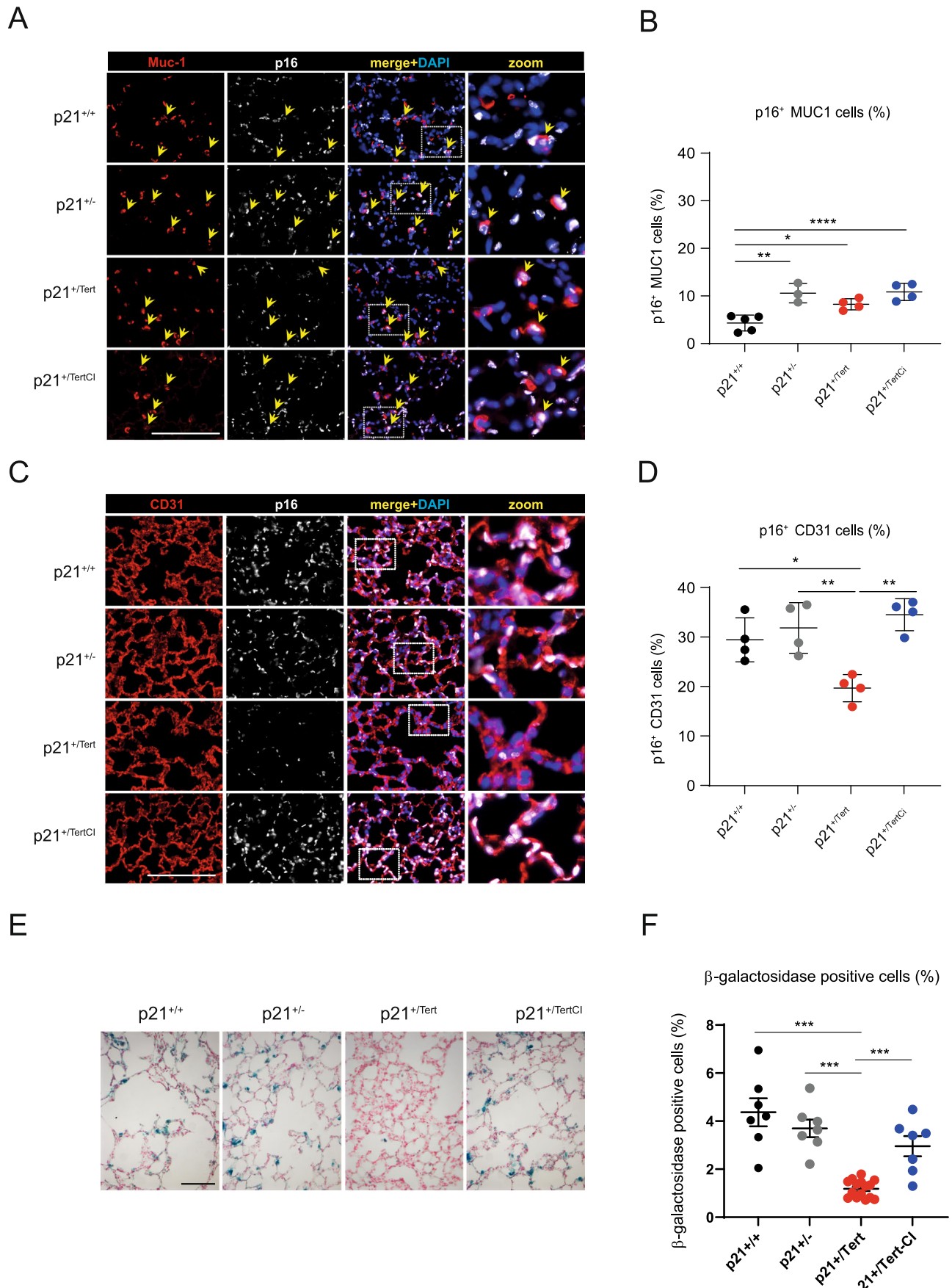

**Figure 5.  Endothelial cell senescence is attenuated in aged p21$^{+/Tert}$ mice.**

(A) Representative micrographs showing immunofluorescence of p16 (white) in lung cells co-stained either with Muc1 (red, a marker of AT2 cells). Arrows indicate cells co-stained for Muc1 and p16. The zoomed areas are indicated by rectangles. Blue rectangle—DAPI nuclear staining. Scale bar—50 μm. (B) Scatter-plot graphs representing the percentage of p16$^+$ Muc1$^+$ cells in the different groups of mice. Data are expressed as individual values per mice and mean ± SEM for groups of mice. (C) Same as (A) except that p16 lung cells (white) are co-stained with CD31 (red, a marker of endothelial cells, lower panel). Scale bar—50 μm. (D) Same as (B) except that the graph represents the percentage of p16$^+$ CD31+ cells. Data are expressed as individual values per mice and mean ± SEM for groups of mice. Data information: *$p < 0.05$, **$p < 0.01$, ****$p < 0.001$ (one way ANOVA with Bonferroni post hoc test). (E) Representative micrographs of SA-beta-Gal staining (blue) in the lungs of aged mice (18-month-old). Red: fast red nuclear staining. Scale bar: 50 μm. (F) Scatter-plot graph showing the percentage of SA-beta-Gal positive cells in each group. ***$p < 0.01$ (one way ANOVA followed by Bonferroni post hoc test). Source data are available online for this figure.

We finally co-marked p21 and CD34 in the lungs of young and old mice (Fig. 8D). The results show that in young mice of all 4 genotypes, p21 labeling is associated with CD34 labeling, in agreement with Tabula Muris Senis showing that *Cdkn1a*$^{p21}$ expression is associated with *Cd34* expression (Fig. EV5). Remarkably, in the old WT, p21$^{+/-}$, and p21$^{+/TertCI}$ mice p21 labeling occurred in the CD34-negative cells, unlike in the old p21$^{+/Tert}$ mice which showed virtually no p21 labeling of the CD34-negative cells. Residual p21 labeling was observed in the CD34+ cells as in the young mice of all genotypes (Fig. 8D). This important observation suggests that maintenance of the pool of putative CD34+ progenitors prevents accumulation of the CD34-negative cells labeled with p21 in the old lung parenchyma.

## p21$^{+/Tert}$ mice exposed to SU5416 under hypoxia are protected against emphysema

Based on our results showing that mTert expression counteracts the age-related decline in capillary density and lung emphysema development presumably by acting on endothelial cells, we asked whether p21$^{+/Tert}$ mice could be protected against lung emphysema primarily originating from a loss of capillary endothelial cells. To this end, we used the SU5416+ hypoxia mouse model in which mice are exposed to the combined treatment with the VEGF receptor inhibitor SU5416 and chronic hypoxic exposure (Shapiro, 2000). SU5416 is an inhibitor of VEGFR2 that causes lung emphysema in rats and mice (Kasahara et al, 2000; Kojonazarov et al, 2019). In recent studies, we reported that mice treated with SU5416 + hypoxia exhibit a huge accumulation of lung senescent endothelial cells, together with a rarefaction in lung capillary density (Born et al, 2023). Here, we found that WT and p21$^{+/-}$ mice as well as p21$^{+/TertCI}$ mice developed airspace enlargement and subsequently lung emphysema when exposed to hypoxia for 21 days with a further aggravation of lung emphysema severity in condition of combined treatment with SU5416 (Fig. 9A–D). In contrast, p21$^{+/Tert}$ mice were protected against lung emphysema development, an effect not found in p21$^{+/TertCI}$ mice (Fig. 9A–D). Importantly, the loss of capillary density induced by the SU5416 + hypoxia exposure (assessed by CD31-stained pulmonary endothelial cells) as well as the increase in p16-positive cells observed in WT, p21$^{+/-}$ and p21$^{+/TertCI}$ mice was not seen in p21$^{+/Tert}$ mice (Fig. 10, upper and lower panel).

Taken together, these results demonstrate that in the model of experimentally induced emphysema, similarly to what occurs during natural aging, *mTert* expressed under the control of the p21 promoter protects the lungs against capillary vessel loss and the development of pulmonary emphysema, primarily by counteracting endothelial cell senescence.

## Discussion

Numerous studies have shown that abnormally short telomeres and senescent cell accumulation are found in the lungs of COPD patients suggesting a direct relationship between short telomeres, senescence and the development of the disease (Alder et al, 2011; Stanley et al, 2014; Houben et al, 2009; Savale et al, 2009; Tsuji et al, 2010). The lung is one of the organs where clinical phenotypes exist in humans when telomeres are shortened beyond a critical point. These conclusions have been strengthened by mouse studies showing that alveolar stem cell failure is a driver of telomere-mediated lung disease (Alder et al, 2015). Collectively these results suggest that DNA damage, cellular senescence, stem cell exhaustion, and likely mitochondrial dysfunction, all induced by accumulation of short telomeres, contribute together to lung functional demise. Because TERT may reverse these processes, TERT-based gene therapy (Martínez and Blasco, 2017) may be clinically beneficial to treat COPD. However, the mechanism by which in vivo TERT expression could prevent the occurrence of emphysema during aging remained to be elucidated.

In this study, we designed a fine-tuned regulatory loop allowing the expression of telomerase under conditions that induce the expression of p21, including the accumulation of critically short telomeres. Importantly this conditional expression of mTert has been introduced in telomerase-positive mice (Tert$^{+/+}$), e.g., with long telomeres underscoring the importance of few critically short telomeres even in WT mice. Although p21 expression is not only dependent on p53, the p21$^{+/Tert}$ mouse provides a unique tool allowing us to evaluate the impact of mTert re-expression on age-related diseases. We exploited the fact that C57BL/6NR mice naturally develop age-related emphysema and mild fibrosis to test in vivo the specific effects of mTert expression (Beaulieu et al, 2021).

Our results indicate that catalytically active mTERT is able to sustain a pool of putative progenitor cells marked with CD34 throughout life, and this correlates with the maintenance of capillary density during aging. Remarkably, the p21 promoter-driven expression of mTert has several related consequences in aging lungs: (1) reduction of the age-dependent accumulation of senescent cells, many of them being endothelial cells, (2) promoting cell proliferation, particularly of the CD34+ endothelial cells, (3) life-long maintenance of the lung capillary density, and (4) attenuation of age-related emphysema and perivascular fibrosis. Of note, we never observed lung tumors in the p21$^{+/Tert}$ mice.

Collectively, these results lead us to propose that maintenance of capillary density in aged mice linked to the sustained proliferation of CD34+ cells could protect lungs against age-related emphysema and perivascular fibrosis. The fact that mTERT could prevent age-

## A

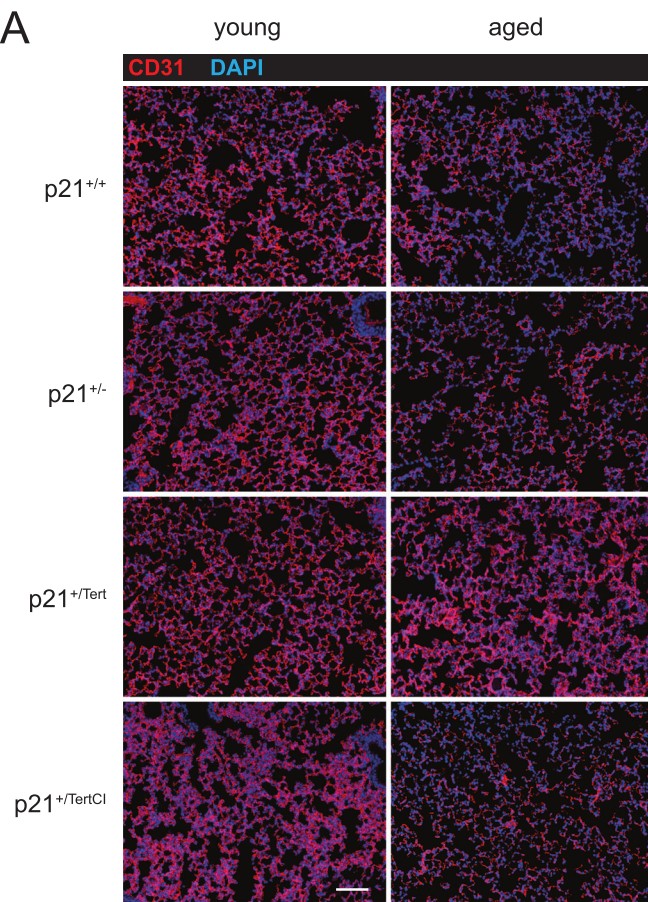

young aged

CD31 DAPI

p21+/+

p21+/−

p21+/Tert

p21+/TertCI

## B

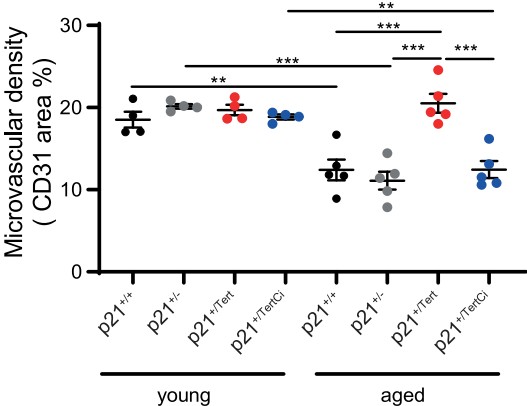

Microvascular Endothelial cells (CD31+) density

young aged

**Figure 6. mTert expression in p21-positive cells counteracts age-related decline in capillary density.**

(A) Representative micrographs showing immunofluorescence of CD31 (red, a marker of endothelial cells). Blue—DAPI nuclear staining. Scale bar—50 μm. (B) Scatter-plot graph showing microvascular density in different groups of young (4 months) and old (18 months) mice. Data are expressed as individual values per mice and mean ± SEM for groups of mice. **$p < 0.01$, ***$p < 0.001$ (one way ANOVA with Bonferroni post hoc test). Source data are available online for this figure.

related emphysema by promoting microvasculature is supported by previous studies showing a tight relationship between vascular density, endothelial cells function and emphysema (Grunewald et al, 2021; Cordasco et al, 1968; Kasahara et al, 2000; Kasahara et al, 2001; Hisata et al, 2021). It was recently shown that VEGF signaling was greatly reduced in multiple organs during aging, and that supplying VEGF protected age-related capillary loss (Grunewald et al, 2021). Conversely, VEGF receptor blockade in rats and mice has become a common experimental model of pulmonary emphysema when combined with chronic exposure to hypoxia (Kasahara et al, 2000; Kojonazarov et al, 2019). In a recent study, we demonstrated that the combination of inhibition of vascular endothelial growth factor receptor 2 (VEGFR2/KDR) by SU5416 and chronic hypoxia was associated with the accumulation of senescent pulmonary vascular endothelial cells as well as to rarefaction of the pulmonary capillaries in young mice (Born et al, 2023). Here we show that young p21+/Tert mice are protected against capillary vessel loss and subsequent development of pulmonary emphysema when treated with SU5416 and hypoxia. Taken together, these results highlight that mTERT may protect against lung emphysema dependent on aging-related endothelial cell senescence and loss of VEGF signaling.

We found that inactivation of the mTert catalytic activity abolishes the protection, thus pointing out the crucial role of mTERT canonical activity in sustaining lung function during aging. This is in agreement with studies indicating that short telomeres are important for the predisposition to lung diseases in patients with mutations affecting telomere elongation by telomerase (Stanley et al, 2014). Interestingly, we detected a correlation between the presence of very short telomeres (<400 bp) and cellular senescence in the lung. We thus assume that induction of the mTert expression from activated p21 promoter might repair damaged telomeres. Indeed, it has been previously reported that the DNA damage response at telomeres was contributing to lung aging and COPD (Birch et al, 2015).

Although the importance of mTERT catalytic activity is well demonstrated in our study, we have limited evidence for the contribution of mTERT non-canonical functions in lung protection. It was previously reported that a catalytically inactive mTERT was able to cause proliferation of hair follicle (Sarin et al, 2005), skin basal keratinocytes (Choi et al, 2008) and kidney podocytes (Shkreli et al, 2011). In particular, transient overexpression of catalytically inactive mTert (i-TERTci mouse model) triggers the clonal expansion of podocyte progenitor cells, however these cells are unable to differentiate when overexpression of mTERTci is maintained inducing lethal nephropathy (Shkreli et al, 2011; Montandon et al, 2022). In our p21+/TertCI model the role of TERT non-canonical function is difficult to reveal because TERT catalytic activity is essential for lung protection. Still, our finding that both mTERT and mTERTCI reduced accumulation of the p21-positive cells in the lungs of aged mice points to the involvement of the mTERT non-canonical function. One tantalizing mechanistic explanation of this effect is that mTERT could stabilize MYC protein (Koh et al, 2015), a known transcriptional repressor of the *Cdkn1a* (p21) gene (Gartel et al, 2001; García-Gutiérrez et al, 2019). Incidentally, it has been recently reported that GSK3β-mediated MYC degradation is required for irreversible cell cycle exit associated with senescence (Afifi et al, 2023), opening a possibility that TERT may counteract senescence by stabilizing MYC in

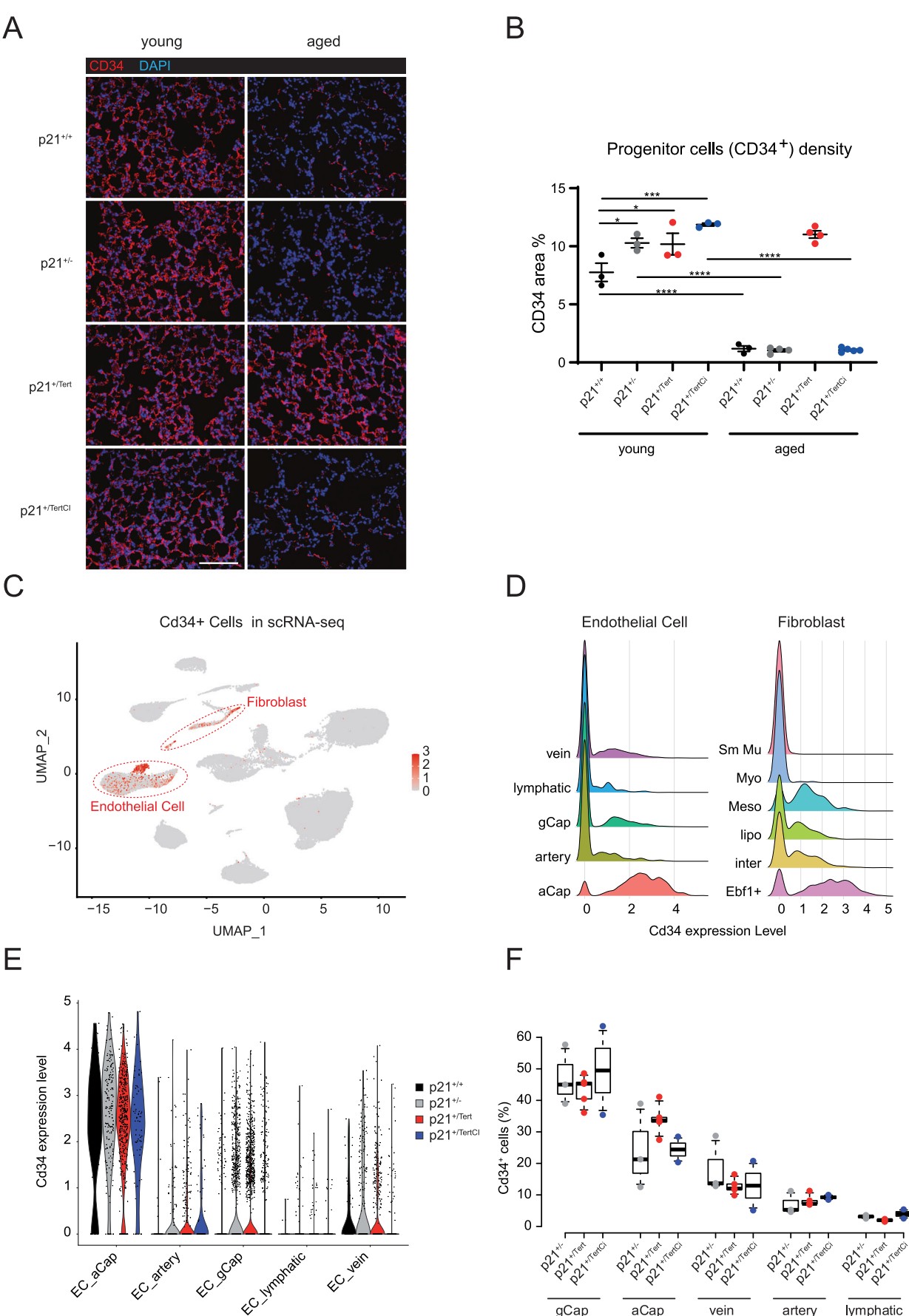

Figure 7.   mTert expression maintains a high number of CD34⁺ cells in the lungs of old mice.

(A) Representative micrographs showing immunofluorescence of CD34 (red). DAPI nuclear staining (blue). Bar—50 µm. (B) Scatter-plot graph showing lung area of CD34 expression in different groups of young (4 months) and old (18 months) mice. Data are expressed as individual values per mice and mean ± SEM for groups of mice. $*p < 0.05$, $***p < 0.001$, $****p < 0.0001$ (One way ANOVA with Bonferroni post hoc test). (C) UMAP clustering of lung cells from 1xWT, 3xp21$^{+/-}$, 5xp21$^{+/Tert}$ and 2xp21$^{+/TertCl}$ mice. Cells expressing *Cd34* gene are visualized by red dots (negative cells are in gray). (D) Ridget plot of *Cd34* expression level for each subtype of EC and fibroblast cell types. (E) Violin plots representing the expression (log(counts)) of *Cd34* in EC subtypes for each genotype. (F) Boxplot of the distribution of *Cd34*⁺ cells in each EC subtype relative to the total number of *Cd34*⁺ ECs. Individual mice are represented. p21$^{+/+}$ (×3, gray), p21$^{+/Tert}$ (×5, red) and p21$^{+/TertCl}$ (×2, blue). Source data are available online for this figure.

addition to its accepted role in telomere stabilization. Yet, we did not find that TERT$^{Cl}$ reduced age-related cellular senescence in the lungs assessed using p16 and SA-β-Gal staining. This could be readily explained by the inability of TERT$^{Cl}$ to maintain telomeres, a function that is likely crucial during CD34+ progenitor cells expansion in the oxidative environment of the lung. Moreover, expression of an inactive mTERT probably creates an excess of telomere damage resulting in an induction of p16 (Jacobs and de Lange, 2004). Finally, the fact that TERT$^{Cl}$ reduces age-related accumulation of the p21+ but not of the p16+ cells suggests that these are two different cell populations.

Previous study showed that after pneumonectomy, stimulated pulmonary capillary EC marked with VE-cadherin, VEGFR2, FGFR1, and CD34 produce angiocrine factors stimulating proliferation of EC progenitors supporting alveologenesis (Ding et al, 2011).

A subclass of endothelial cells interacting with AT1 cells and expressing high levels of *Car4*, *Kdr*, and *Cd34* was described as essential for regeneration of pulmonary alveolar endothelium after acute lung injury (Niethamer et al, 2020). We report in this study that CD34+ cells in the p21$^{+/Tert}$ mice are marked by PCNA staining and incorporate BrdU reflecting their ability to proliferate. Of note, these cells are primed to respond to VEGFA due to their high level of VEGF receptor (Niethamer et al, 2020). In the single-cell analysis, we aimed at identifying the nature of the CD34+ endothelial cells and their properties. We found that Cd34 was expressed in cells defined as gCap cells and aerocytes (Gillich et al, 2020). Because aerocytes have been described to rarely proliferate even after injury (Gillich et al, 2020), the question remains about the identity of the proliferating CD34+ cells. Because aerocytes may be generated from gCap during repair (Gillich et al, 2020), one possibility is that gCaps that express *Cd34* at low levels generate aCaps marked by CD34 in p21$^{+/Tert}$ mice. This hypothesis is consistent with the aCap/gCap ratio being higher in the p21$^{+/Tert}$

mice compared to the control mice. Alternatively, since both cell types develop from bipotent progenitors (Gillich et al, 2020), another possibility is that p21-promoter dependent expression of mTert promotes the differentiation of the progenitors into aerocytes. Since we could not reveal the differences in the level of *Cd34* mRNA among the four genotypes by scRNA-seq, in contrast to the CD34 immunostaining, we propose that the level of the CD34 surface protein could be regulated by receptor-mediated endocytosis (Krauter et al, 2001) independently of the *Cd34* gene transcription.

In conclusion, our results support a model in which p21 promoter-dependent expression of m*Tert* preserves putative EC progenitors marked by CD34 in the lungs of old mice, which leads to enhanced regenerative capacity and reduced EC senescence. This would be particularly important in maintaining the capillary density. As a consequence, this will reduce age-related emphysema and perivascular fibrosis. Overall, our results open new avenues for the development of treatments based on the conditional expression of m*Tert* targeting lung endothelial cells to cure lung diseases. Nevertheless, the oncogenic potential of telomerase expression under the control of the p21 promoter will have to be evaluated in a very rigorous way, even if we did not observe in p21$^{+/Tert}$ mice any lung cancer. In the recent years, trials have been made to explore the potential of CD34+ cell therapy for the treatment of different cardiovascular diseases (Prasad et al, 2020). We can anticipate that the long-term maintenance of CD34+ endothelial cells could have therapeutic benefits beyond pulmonary pathologies.

## Methods

### Reagents and tools

See Table 1.

**Table 1.   Reagents and tools.**

| Reagent/resource | Reference or source | Identifier or catalog number |
| --- | --- | --- |
| Experimental models | | |
| C57Bl/6 N (*M. musculus*) | Plateforme KOKI-Ciphe (Marseille) | 1310_mCherry-2A-mTert (p21$^{+/Tert}$ mouse model) |
| C57Bl/6 N (*M. musculus*), littermate controls | Plateforme KOKI-Ciphe (Marseille) | 1310_mCherry-2A-mTert (p21$^{+/+}$ mouse model) |
| C57BL/6J (*M. musculus*) | Plateforme KOKI-Ciphe (Marseille) | B6-Cdkn1a$^{tm2Ciphe}$ (p21$^{+/TertCl}$ mouse model) |
| C57BL/6J (*M. musculus*) | Jackson Laboratory | B6.129S6(Cg)-Cdkn1a$^{tm1led}$/j (p21$^{+/-}$ mouse model) |
| Pulmonary artery smooth muscle (PA-SMC) | Isolated in the CRCM | p21$^{+/+}$, p21$^{+/Tert}$ and p21$^{+/TertCl}$ |

**Table 1.** (continued)

| Recombinant DNA | | |
|---|---|---|
| pCAG-GFP | Addgene | 11150 |
| pCAG-mCherry-2A-mTert | This study | |
| **Antibodies** | | |
| Rabbit p16-INK4 | abbiotec | 250804 |
| Mouse anti-p21 | BD Pharmingen | 556431 |
| Mouse phospho-Histone H2A.X (Ser139) | Millipore | 05-636 |
| IsolectinB4 alexafluor 594 | ThermoFisher | I21413 |
| Rat anti-p21 | Abcam | ab107099 |
| Rabbit anti-CD31 | Abcam | ab182981 |
| Rabbit anti-MUC1 | Abcam | ab109185 |
| Rabbit anti-CD34 | Abcam | ab81289 |
| Mouse anti-CDKN2A/p16INK4a | Abcam | ab54210 |
| Mouse anti-PCNA | Abcam | ab29 |
| Mouse anti-BrdU | Abcam | ab8152 |
| Recombinant anti-p21 antibody [EPR3993] | Abcam | ab109199 |
| Beta-actin (from hybridoma) | Sigma | A1978 |
| Rabbit anti-53BP1 | Novus | NB100 |
| Goat Anti-Mouse Immunoglobulins/HRP | Dako | P0447 |
| Goat Anti-Rabbit Immunoglobulins/HRP | Dako | P0448 |
| Donkey polyclonal anti Mouse Alexa555 | ThermoFisher | A31570 |
| Donkey anti-Rabbit IgG Alexa Fluor 647 | ThermoFisher | A31573 |
| Mouse anti-8-oxo-dG (15A3) | R&D Système Bio-techne | #4354-MC-O50 |
| Goat anti-Rabbit IgG Alexa Fluor 647 | ThermoFisher | A21074 |
| Goat anti-Rabbit IgG Alexa Fluor 555 | ThermoFisher | A32732 |
| Goat anti-mouse IgG Alexa Fluor 647 | ThermoFisher | A21054 |
| Goat anti-mouse IgG Alexa Fluor 555 | ThermoFisher | A32727 |
| Goat anti-rat IgG Alexa Fluor 555 | ThermoFisher | A21434 |
| **Oligonucleotides and sequence-based reagents** | | |
| Genotyping PCR primer | This study | See Genotyping |
| Primers for qPCR transcript | This study | See qPCR |
| Oligonucleotides and primers used in TRAP assay and TeSLA | This study | See TRAP and TeSLA assay |
| Telomeric PNA-probe - Alexa488 | PANAGENE | F1008 |
| **Chemicals, enzymes, and other reagents** | | |
| Doxorubicin | Merck | D1515 |
| DAPI | Sigma | D9542 |
| DMEM (Dulbecco Modified Eagle Medium, High Glucose) | Gibco/Thermo Fischer Scientific | 10938-025 |
| Penicillin/streptomycin | Gibco | 15140-122 |
| GlutaMax | Life Technologies | 35050061 |
| NEAA | Life Technologies | 11140050 |
| Trypsin (10×) | Life Technology | 15400-054 |
| Sodium pyruvate | Gibco | 11360070 |
| DMSO | Sigma | D2640 |
| Blocking Reagent | Roche | 11095176001 |
| VectaShield containing DAPI | Vector Lab | H-1200 |

    

**Table 1.** (continued)

| | | |
|---|---|---|
| Picro Sirius Red Stain Kit | Abcam | ab150681 |
| 4% paraformaldehyde | Biosharp | BL539A |
| Antigen retrieval buffer | Abcam | ab93684 |
| Collagenase/dispase | Roche | 11097113001 |
| Nylon cell strainer | Fisher Scientific | 11873402 |
| Superscript IV Reverse Transcriptase | Invitrogen | 18090050 |
| SYBR Green master mix | Takara bio | 639676 |
| Protease inhibitors cOmplete | Roche | 11697498001 |
| RNase inhibitor | Applied Biosystems | N8080119 |
| FailSafe polymerase mix | Epicenter | FSE5101K |
| Red blood cell lysis | Invitrogen | 00-4333-57 |
| Software | | |
| M3vision | Biospace Lab | |
| EVOS M5000 Imaging System | ThermoFisher | |
| Nikon NIS element AI software | Nikon | |
| OMERO | Open Microscopy Environment | |
| NIH ImageJ | https://imagej-nih-gov.insb.bib.cnrs.fr/ij/ | |
| Image Studio Lite ver 5.2 | LI-COR | |
| TeSLA Quant software | Lai et al (2017) | |
| GraphPad Prism 9 Software | GraphPad | |
| Statistica 13.0 software package | StatSoft/Dell | |
| v2.3 Cell Ranger pipeline | 10× Genomics | |
| perCellQCMetrics | Scatter package | |
| scran | Bioconductor 3.18 | |
| Seurat v4.1.0 | https://satijalab.org/seurat/ | |
| ZEN 3.0 (blue edition) | ZEISS group | |
| Other | | |
| Photon-IMAGER | Biospace Lab | |
| RS2000 Irradiator | Radsources | |
| Nikon AX | Nikon | |
| Nikon Ti-E | Nikon | |
| Ventilated chamber | Biospherix | |
| Axio Imager M2 imaging microscope | Zeiss | |
| ChemiDoc MP imaging system | Bio-Rad | |
| Bioanalyzer | Agilent | |
| NovaSeq sequencer | Illumina | |
| ZEISS Axioscan 7 Microscope Slide Scanner | ZEISS group | |
| RNeasy kit | Qiagen | 74104 |
| Immunodetection kit | Vector Lab | BMK-2202 |
| BrdU immunohistochemistry kit | Abcam | ab125306 |
| Pierce™ 660nm Protein Assay | ThermoFisher | 22660 |
| Single cell V2 reagent kit | 10× Genomics | PN-120237 |

**Table 1.** (continued)

**Genotyping primer sequences**

| Name | Sequences | PCR size product |
| --- | --- | --- |
| 1310_ScES_01 F | CTGAATGAACTGCAGGACGA | |
| 1310_ScES_01_R | CTTGCCTATATTGCTCAAGG | |
| WT p21 F | GCTGAACTCAACACCCACCT | 435 |
| WT p21 R | GCAGCAGGGCAGAGGAAGTA | |
| mutant p21-mCherry-2A-Tert F | GGACCTCTGAGGACAGCCCAAA | 503 |
| mutant p21-mCherry-2A-Tert R | GCAGCAGGGCAGAGGAAGTA | |
| mutant p21-mCherry-2A-Tert-Cl F | GGACCTCTGAGGACAGCCCAAA | 520 |
| mutant p21-mCherry-2A-Tert-Cl R | GCAGCAGGGCAGAGGAAGTA | |
| B6.129S6(Cg)-*Cdkn1a*<sup>tm1Led</sup>/J F | GTTGTCCTCGCCCTCATCTA | 240 |
| B6.129S6(Cg)-*Cdkn1a*<sup>tm1Led</sup>/J R | GCCTATGTTGGGAAACCAGA | |
| B6.129S6(Cg)-*Cdkn1a*<sup>tm1Led</sup>/J F | GTTGTCCTCGCCCTCATCTA | 447 |
| B6.129S6(Cg)-*Cdkn1a*<sup>tm1Led</sup>/J R | CTGTCCATCTGCACGAGACTA | |

**Primers for qPCR transcript**

| Name | Sequences |
| --- | --- |
| mTert-2253 F | AGCCAAGGCCAAGTCCACAA |
| mTert-2399 R | AGAGATGCTCTGCTCGATGACA |
| mCherry-2A F2 | AGCAGGAGATGTTGAAGAAAACCC |
| mCherry-2A-mTert-5′ R2 | GGCCACACCTCCCGGTATC |
| mActb-90 F | ACACCCGCCACCAGTTCG |
| mActb-283 R | GGCCTCGTCACCCACATAGG |
| mTert-2582 F | GCGGGATGGGTTGCTTTTACG |
| mTert-2688 R | ACTCAGGAACGCCATGGACC |
| mCherry-2A F | GAGCTGTACAAGGGAGCCACG |
| mTert-2A R | CGAGGAGCGCGGGTCATC |
| m_p21_F1 | GGAATTGGAGTCAGGCGCAG |
| m_p21_R1 | CCGAAGAGACAACGGCACAC |
| m_p21_F4 | AGCAGCCGAGAGGTGTGAG |
| m_p21_R4 | CATGGTGCCTGTGGCTCTG |
| mCherry_R1 | TGTGCACCTTGAAGCGCATG |

**Oligonucleotides and primers used in TRAP assay and TeSLA**

| Name | Sequences | Modification |
| --- | --- | --- |
| TS, telomerase extension substrate | 5′-AATCCGTCGAGCAGAGTT | 5′FAM mod. |
| ACX, reverse amplification primer | 5′-GCGCGGCTTACCCTTACCCTTACCCTAACC | |
| TSNT, internal competitor oligo | 5′-AATCCGTCGAGCAGAGTTAAAAGGCCGAGAAGCGAT | |
| NT, primer | 5′-ATCGCTTCTCGGCCTTTT | |
| Oligos for telomeric DNA ligation | | |
| TeSLA Telo 1 | ACTG GCC ACG TGT TTT GAT CGA CCC TAA C | |
| TeSLA Telo 2 | ACTG GCC ACG TGT TTT GAT CGA TAA CCC T | |
| TeSLA Telo 3 | ACTG GCC ACG TGT TTT GAT CGA CCT AAC C | |
| TeSLA Telo 4 | ACTG GCC ACG TGT TTT GAT CGA CTA ACC C | |

**Table 1.** (continued)

| | | |
|---|---|---|
| TeSLA Telo 5 | ACTG GCC ACG TGT TTT GAT CGA AAC CCT A | |
| TeSLA Telo 6 | ACTG GCC ACG TGT TTT GAT CGA ACC CTA A | |
| Oligos for subtelomeric DNA ligation | | |
| TeSLA ADR1 C3S | GGT TAC TTT GTA AGC CTG TCˆ | C3 spacer at 3' end (avoid DNA ligation) |
| TeSLA P22 TA | *TA GAC AGG CTT ACA AAG TAA CCA TGG TGG AGA ATT CTG TCG TCT TCA CGC TAC ATTˆ | C3 spacer at 3' end (avoid DNA ligation) + phosphorylation at 5' end (facilitate DNA ligation) |
| TeSLA P22 AT | *AT GAC AGG CTT ACA AAG TAA CCA TGG TGG AGA ATT CTG TCG TCT TCA CGC TAC ATTˆ | C3 spacer at 3' end (avoid DNA ligation) + phosphorylation at 5' end (facilitate DNA ligation) |
| Oligos for TeSLA PCR | | |
| TeSLA AP | TGT AGC GTG AAG ACG ACA GAA | |
| TeSLA TP | TGG CCA CGT GTT TTG ATC GA | |

## Generation of mCherry-2A-Tert and mCherry-2A-Tert^CI mice

Mouse lines generated in this study are deposited at Ciphe, Marseille, France. Generation of the p21^+/Tert mouse model. A 6.4 kb genomic fragment encompassing coding exons 2 and 3 of the *Cdkn1a* gene was isolated from a BAC clone of B6 origin (clone no RP23-412O16; http://www.lifesciences.sourcebioscience.com) and subcloned into the *Not* I site of pBluescript II. The genomic sequence containing the homologous arms was checked by DNA sequencing. A synthetic gene encoding the mCherry, 2A peptide, and the 5' coding region of m*Tert* (beyond the *Sac* II site) was constructed by gene synthesis (By DNA2.0 Inc) to produce the *mCherry-2A-mTert^SacII* cassette. Using ET recombination, the synthetic *mCherry-2A-mTert^SacII* cassette was introduced in the 5' coding region of the *Cdkn1a* gene replacing the start codon. The full-length m*Tert* was next introduced in the *mCherry-2A-mTert^SacII* truncated cassette by inserting into the targeting vector a synthetic m*Tert* (*Sac* II-*EcoR* I) fragment restoring the full-length mTert to give rise to the complete *mCherry-2A-mTert* cassette. A self-deleter NeoR cassette was further introduced in the *EcoR* I site of the targeting vector immediately after the mTert sequence. A cassette coding for the diphtheria toxin fragment A expression cassette was finally introduced in the *Not* I site of the targeting vector. All the elements of the final targeting are shown in the Appendix Fig. S1.

JM8.F6 C57BL/6N ES cells were electroporated with the targeting vector linearized with *Fse*I (Pettitt et al, 2009). The scheme of the introduction of the *mCherry-2A-mTert-Neo^R* cassette in place of the ATG codon of Cdkn1a is shown in the Appendix Fig. S1A,B. After selection with G418, ES cell clones were screened for proper homologous recombination. The correct integration of the cassette was verified by Southern blot with a *Cdkn1a* probe (Appendix Fig. S1C) and by long range PCR (Appendix Fig. S1D). The primers used to check the correct insertion of the cassette were: 1310_ScES_01 Fwd: 5' CTGAAT-GAACTGCAGGACGA; 1310_ScES_01_Rev: 5' CTTGCCTATATTGCT CAAGG. To ensure that adventitious non-homologous recombination events had not occurred in the selected ES clones, a Southern blot was performed with a mCherry probe (Appendix Fig. S1E).

Properly recombined ES cells were injected into Balb/c blastocysts. Germline transmission led to the self-excision of the loxP-Cre-NeoR-loxP cassette in male germinal cells. p21-mCherry-2A-mTert mice (p21^+/Tert) were identified by PCR of tail DNA. The primers used for genotyping were as follows: sense-WT 5'-GCTG AACTCAACACCCACCT-3', sense-p21-Tert 5'-GGACCTCTGA GGACAGCCCAAA-3' and antisense 5'-GCAGCAGGGCAGAGG AAGTA-3'. The resulting PCR products were 435 (WT) and 520 (p21^+/Tert) base pairs long. The proteins produced by the mRNA mcherry-2A-mTert polycistronic mRNA are shown in Appendix Fig. S1F. In the final construct, Cdkn1a^p21 protein is no further produced from the mutated allele because of the substitution of the *Cdkn1a* [ATG] codon by the cassette and the presence of a STOP codon at the end of m*Tert*.

As control mice, we developed p21^+/+ littermates and p21^+/- mice that were obtained by crossing the p21 homozygous knockout strain B6.129S6(Cg)-Cdkn1a^tm1Ied (from JAX laboratories) with isogenic p21^+/+ mice. In the p21^+/- mouse, a neo cassette replaces exon 2 of *Cdkn1a*. Homozygotes are viable, fertile, and of normal size.

p21^+/TertCI mice were constructed with exactly the same targeting vector used to construct p21^+/Tert except that the codon GAT in m*Tert* coding for D702 has been changed into GCA coding for A. Primers to genotype p21^+/Tert and p21^+/TertCI mice are indicated in the reagents and tools table (Table 1).

## Ethical statement

Mice were used according to institutional guidelines, which complied with national and international regulations. All animals received care according to institutional guidelines, and all experiments were approved by the Institutional Ethics committee number 16, Paris, France (licence number 16-093). Mice were bred and maintained in specific-pathogen-free conditions with sterilized food and water provided ad libitum and were maintained on a 12 h light and 12 h dark cycle in accordance with institutional guidelines.

## Verification of mTERT activity

To check that mTERT expressed from the *mCherry-2A-mTert* can produce active telomerase, the *mCherry-2A-mTert* was amplified by

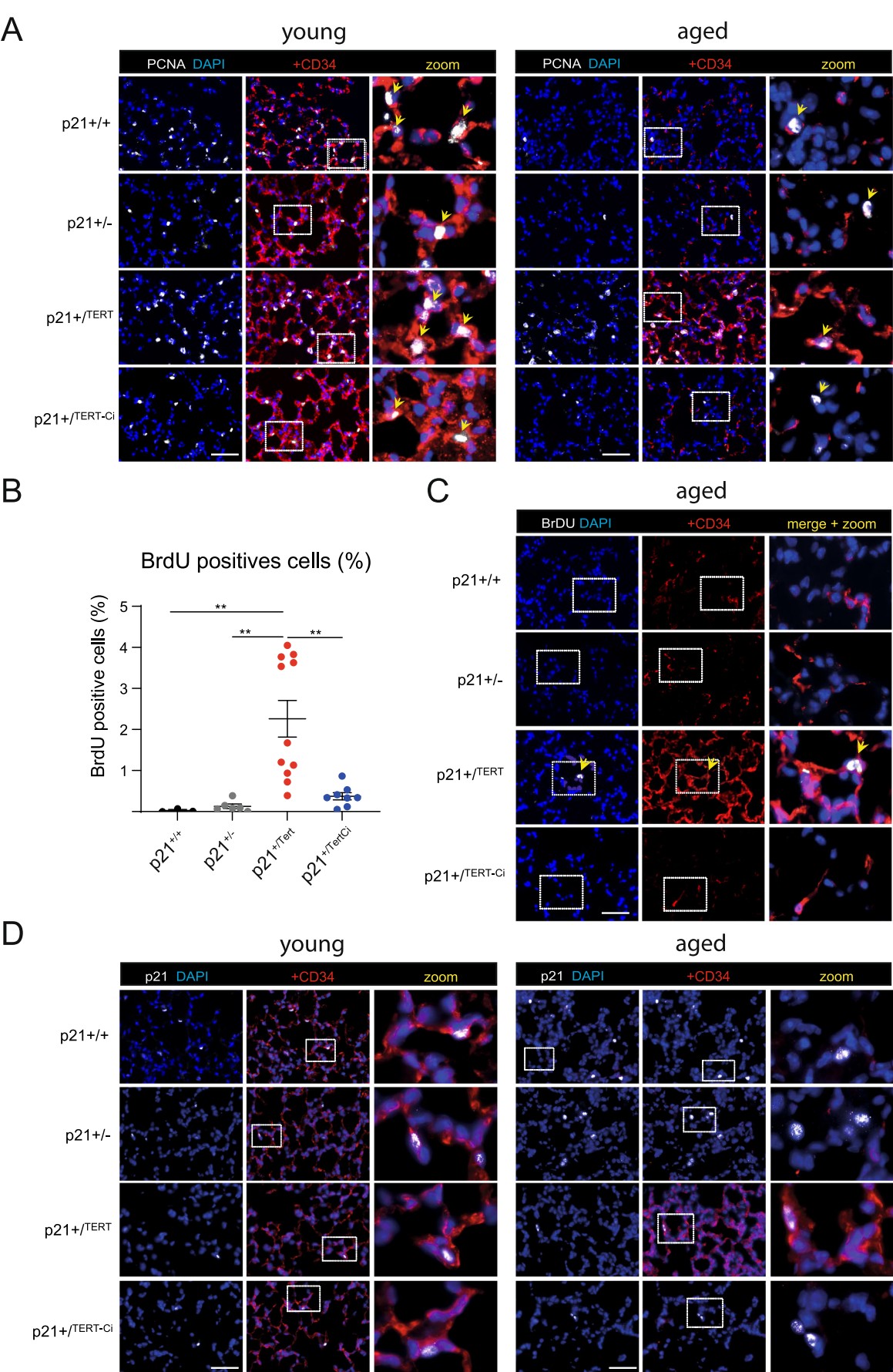

◄ **Figure 8.  Lung CD34$^+$ cells preserve their capacity to proliferate in aged p21$^{+/Tert}$ mice.**

Representative micrographs of mice lung showing expression of CD34 (red) in lung of young and old mice of all genotypes co-stained with (**A**) anti-PCNA antibody (white, proliferating cells nuclear antigen), (**C**) anti-BrdU antibody (white), or (**D**) anti-p21 antibody (white). (**A**) Arrows indicate proliferating cells identified by nuclear PCNA immunofluorescence. The majority of proliferating cells were also co-stained with CD34. (**B**) Scatter-plot graph showing percentage of BrdU-positive cells in different types of mice. Data are expressed as individual values per mice and mean ± SEM for group of mice. **$p < 0.01$, (one way ANOVA with Bonferroni post hoc test). (**C**) Eighteen-month-old mice were injected with BrdU intra-peritoneally 24 h before organ sampling. Arrows indicate proliferating CD34$^+$ cells incorporating BrdU. (**D**) In young mice of all genotypes the majority of p21$^+$ cells are also co-stained with CD34. Among old mice, p21$^+$ CD34$^+$ cells can be detected only in p21$^{+/Tert}$ mice. Data information: Blue—DAPI nuclear staining. The zoomed areas are indicated by rectangle. Scale bar—50 μm. Source data are available online for this figure.

PCR (S 5'-atatat<u>gaatt</u>catggtgagcaagggcgaggaggata; AS 5'- ata-tatgcggccgcttagtccaaaatggtctgaaagtct) from the targeting vector and cloned into the *EcoR* I/*Not* I sites of pCAG-GFP. In the resulting vector, the *mCherry-2A-mTert* cassette is expressed under the control of the strong constitutive CAG promoter. The resulting vector was transfected in the m*Tert*$^{-/-}$ mouse ES cells (provided by Lea Harrington, Montreal). Prior to transfection, the Neo$^R$ cassette in the m*Tert*$^{-/-}$ ES cells was disrupted by CRISPR/CAS9 since the m*Tert*$^{-/-}$ ES cells and the electroporated plasmid shared the same antibiotic resistance. The presence of telomerase activity was analyzed in the cell extracts using the Telomere Repeat Amplification Protocol (TRAP) (see below). We could detect robust telomerase activity in mouse m*Tert*$^{-/-}$ ES cells transfected by the construct pCAG-*mCherry-2A-mTert* (Fig. 1B).

## Fluorescence imaging of mice and organs

To demonstrate in vivo the expression of the KI *Cdkn1a* allele in response to p21 promoter activation, littermate p21$^{+/+}$ and p21$^{+/Tert}$ mice were exposed either to a whole-body ionizing radiation or to doxorubicin which both activate the p53-p21 axis of the DNA damage response (DDR). For irradiation experiments, p21$^{+/+}$ and p21$^{+/Tert}$ mice were imaged (time zero) and then exposed to 1.5-Gy of total body irradiation (RX, RS2000 Irradiator, Radsources). Non-invasive whole-body fluorescence (FLI) was determined at the indicated times post-treatment. Imaging was performed using a Photon-IMAGER (Biospace Lab), and mice were anesthetized with 3% vetflurane through an O$_2$ flow modulator for 5 min and then the image was acquired (4-s exposure, excitation = 545 nm, background = 495 nm, emission filter cut off = 615 nm). Corresponding color-scale photographs and color fluorescent images were superimposed and analyzed with M3vision (Biospace Lab) image analysis software.

For doxorubicin treatments, the p21$^{+/+}$ and p21$^{+/Tert}$ mice were injected i.p. with doxorubicin (DXR) (20 mg/kg). Doxorubicin has been shown to activate p21 promoter to high levels in the liver and kidneys (Tinkum et al, 2011). Twenty-four hours after DXR injection, mice were then sacrificed and organs were rapidly removed and imaged within 15 min of sacrifice (4-s exposure). After that, proteins and RNA were extracted from kidneys and liver for immunoblot (p21) and RT-qPCR (mTert) analyses, respectively.

## Primary pulmonary artery smooth muscle cells and culture

Pulmonary artery smooth muscle (PA-SMC) were extracted and cultured as already described (Houssaini et al., 2018). Cells were cultured at 37 °C, 5% O$_2$, in DMEM (High glucose, -Pyruvate, Gibco) supplemented with 10%(v/v) decomplemented FBS and 1%

(v/v) Penicillin-Streptomycin. At each passage, cells were counted and 50,000 cells were put back in culture in a 25 cm$^2$ flask and cultured in DMEM supplemented with FBS. Representative images of cells in culture were obtained using EVOS M5000 Imaging System (ThermoFisher) on p21$^{+/+}$ and p21$^{+/Tert}$ cells 24 h after plating on a 24-well plate.

## Telomere dysfunction Induced Foci (TIFs)

Paraffin–embedded lung tissue sections (5 μm wide) were dewaxed in xylene (2 × 5 min), re-hydrated in a graded ethanol series (5 min each) and PBS (5 min). Epitope demasking was performed in 10 mM NaCitrate, 0.05% Tween-20, pH=6.0, at 95 °C for 30 min, upon which the slides were rinsed in water, and tissue sections dehydrated in 96% ethanol, and air dried. Telomeric PNA-probe (Alexa488-OO-(TTAGGG)$_3$, from PANAGENE) was diluted in hybridization buffer (70% formamide, 10 mM Tris, pH = 7.2, 1% Blocking Reagent (ROCHE, 11095176001; stock solution: 10% (w/v) Blocking Reagent in 0.1 M Maleic acid, 0.15 M NaCl, pH = 7.5) at a final concentration of 0.5 mM. The probe was then denatured 5 min at 95 °C, and put on ice (2 min). 100 mL of denatured probe was then put on the tissue sections, under sealed coverslip, and probe and target were denatured a further 4 min at 80 °C, on a hot plate. Hybridization was in a moist chamber, overnight, in the dark, at room temperature (RT). All further incubations and washes were also performed in the dark.

Post-hybridization washes were in 70% formamide, 10 mM Tris, pH = 7.2, 30 min, and 50 mM Tris, 150 mM NaCl, 0.05% Tween-20, pH7.5, 15 min at RT. Slides were rinsed in PBS, and blocked in 4% BSA in PBS, 0.1% Tween-20 for 1 h at RT. Primary antibody incubation (Rabbit anti-53BP1, Novus, NB100) was 1/300 in 4% BSA in PBS, 0.1% Tween-20, overnight at 4 °C. Washes were in PBST (PBS, 0.1% Tween-20), 30 min, RT. Secondary antibody incubation (Donkey anti-Rabbit IgG Alexa Fluor 647, A-31573, THERMO FISHER SCIENTIFIC) was diluted 1/1000 in 4% BSA in PBS, 0.1% Tween-20, 2 h at 37 °C. Washes were in PBST, 30 min, RT. Slides were finally rinsed in PBS and mounted in VectaShield containing DAPI (H-1200, VECTOR LABORATORIES).

All images were collected with a Nikon AX on a Nikon Ti-E inverted equipped with PLAN APO LBDA 60x. DAPI, alexa fluor 488 and alexa fluor 647 were excited with the 405, 488 and 640 lines from a LU-N4 LASER UNIT. Images were acquired and analyzed by the CRCM DNA Damage Platform and Imaging facility (DAPI) using Nikon NIS element AI software and quantified in 3D using bright spot detection and colocalization module. Z-series are displayed in figures as maximum z-projections generated by OMERO.

Levels of 8-hydroxy-2'-deoxyguanosine (8-oxo-dG) within DNA was monitored with the Anti-8-oxo-dG (15A3) antibody (Bio-

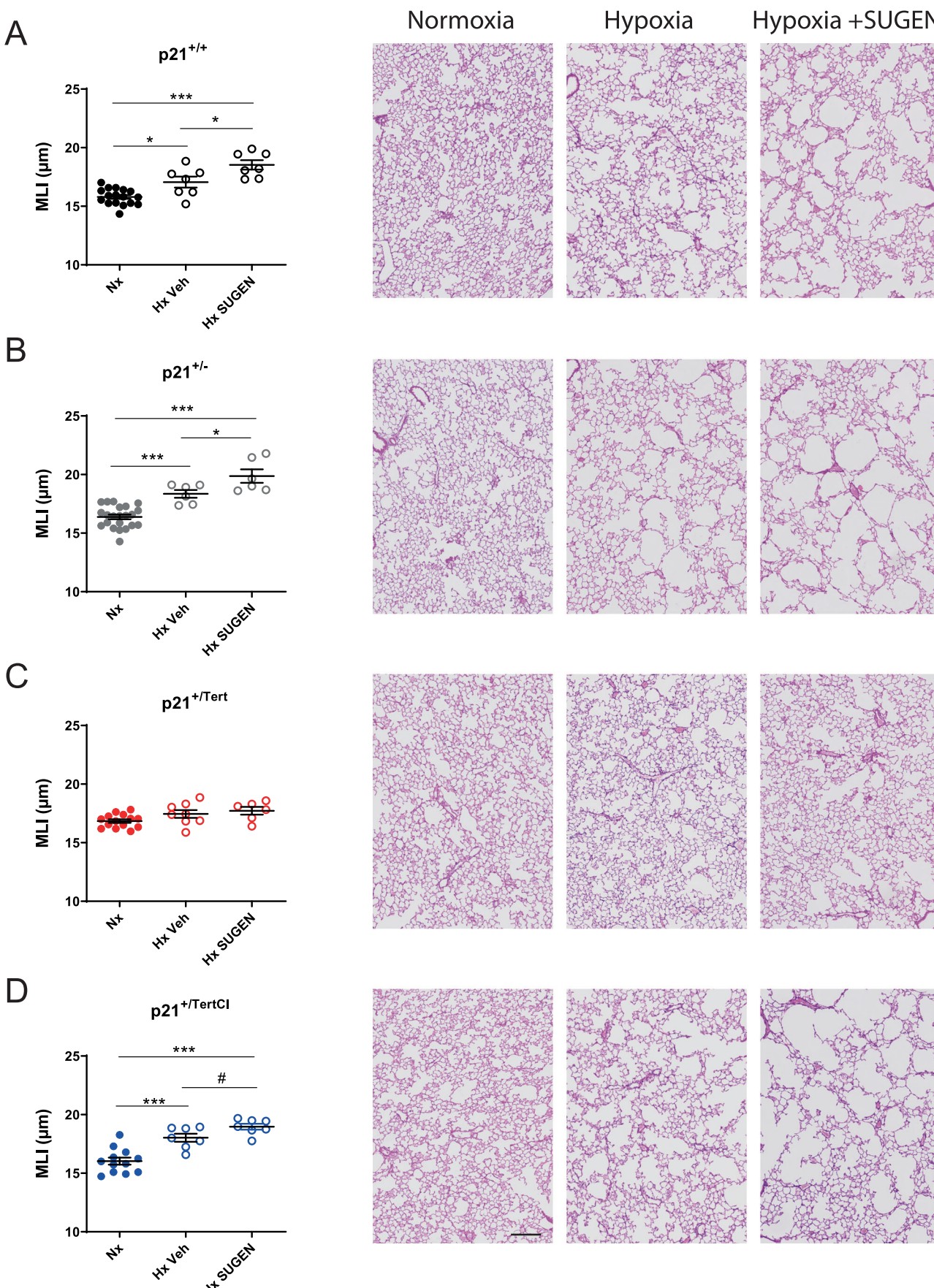

**Figure 9.** **p21$^{+/Tert}$ mice exposed to SU5416 plus hypoxia are protected against emphysema.**

(A–D) Mice of the indicated genotypes were simultaneously treated with SU5416 and exposed to normoxia or chronic hypoxia (21 days). Lung emphysema as assessed by measurement of the mean linear intercept (MLI) in hematoxylin and eosin-stained lung sections. Data are expressed as individual values per mice and mean ± SEM for group of mice. *p < 0.05, ***p < 0.001 for comparison between groups as indicated (one-way ANOVA) followed by Tukey post hoc test). Scale bar—200 µm. Source data are available online for this figure.

techne, Catalog #: 4354-MC-050) following the instruction of the technical note provided by the supplier.

## Animal studies and lung tissue analysis

Groups of mice from the four genotypes were prepared in order to sacrifice 4-month-old mice and 18-month-old mice. To reach the final mice number per group and per genotype, this experiment was reproduced three individual times, on three independent cohorts. For p21$^{+/Tert-CI}$ mice, an independent cohort of p21$^{+/Tert}$ and p21$^{+/Tert-CI}$ mice was prepared.

A subgroup of mice received an intraperitoneal injection of 5-bromo-2'-deoxyuridine (ab142567, abcam) diluted in PBS at 25 mg/mL. Mice received 100 mg/kg BrdU 24 h prior to sacrifice. Mice were anesthetized with an intraperitoneal injection of ketamine (60 mg/kg) and xylazine (10 mg/kg). After mice sacrifice, three lobes of the right lung were quickly removed and immediately snap-frozen in liquid nitrogen then stored at −80 °C for biological measurements. Genomic DNA from the right lung of each animal was used for TeSLA assay. The last lobe of the right lung was fixed with 2% formaldehyde (Sigma) and 0.2% glutaraldehyde (Sigma) for 45 min. Then, lungs were washed with PBS and stained overnight in a titrated pH 6 solution containing 40 mM citric acid, 150 mM NaCl, 2 mM MgCl$_2$, 5 mM potassium ferrocyanide, and 1 mg/ml X-Gal at 37 °C. Stained lobes were then imbedded in paraffin and 5 µm-thick sections were cut. Lung sections were deparaffinized using xylene and a graded series of ethanol and then nuclei were stained with neutral red. 10 fields per section were acquired at an overall magnification of 500. The number of beta-galactosidase stained cells and expressed as a % of total number of cells per field.

The left lungs were fixed by intratracheal infusion of 4% paraformaldehyde aqueous solution (Sigma) at a trans-pleural pressure of 30 cm H$_2$O and processed for paraffin embedding. For morphometry studies, 5 µm-thick sagittal sections along the greatest axis of the left lung were cut in a systematic manner to allow immunostaining. Lung emphysema was measured using mean linear intercept methods on hematoxylin–eosin–saffron (HES) coloration. Briefly 20 fields/animal light microscope fields at an overall magnification of 500, were overlapped with a 42-point, 21-line eyepiece according to a systematic sampling method from a random starting point. To correct area values for shrinkage associated with fixation and paraffin processing, we used a factor of 1.22, calculated during a previous study (Houssaini et al, 2018). In addition, MLI measurement was performed by an independent operator by using semi-automated measurement of MLI (Crowley et al, 2019). Both methods (manual and semi-automated) performed by two independent observers provided the same results.

Lung fibrosis was quantified on lung sections stained with Sirius Red (Picro Sirius Red Stain Kit, ab150681, Abcam, Cambridge, UK) using the modified Ashcroft scale (Hübner et al, 2008). Briefly, the

lungs were scanned microscopically with a 20-fold objective, which allowed the evaluation of fine structures while also providing a sufficiently broad view. The entire section was examined by inspecting each field in a raster pattern. Areas with dominating tracheal or bronchial tissue were omitted. The grades were summarized and divided by the number of fields to obtain a fibrotic index for the lung.

## Exposure to SU5416+ hypoxia

Mice were exposed to chronic hypoxia (9% O$_2$) in a ventilated chamber (Biospherix, New York, NY) as previously described (Born et al, 2023). Mice subjected to SU5416/hypoxia received an intraperitoneal injection of Sugen, 20 mg/kg (Sigma, St Quentin-Fallavier, France) once a week during a 3-week period of hypoxia exposure.

## Immunofluorescence

All antibodies used in this study are described in the reagents and tools table (Table 1).

Paraffin-embedded sections of lung were deparaffinized using xylene and a graded series of ethanol dilutions then processed for epitope retrieval using citrate buffer (0.01 M, pH 6; 90 °C, 20 min). For CD31 epitope retrieval, we used Antigen retrieval buffer (100× Tris-EDTA buffer, pH9.0, ab93684, Abcam) according to the manufacturer's instructions. For nuclear immunolabeling, tissues were permeabilized with 0.1% Triton X-100 in PBS for 10 min. Saturation was achieved using Dako antibody diluents with 10% goat serum. For immunolabeling with primary antibody produced in mouse we used the M.O.M. (mouse on mouse) immunodetection kit, basic (ref. BMK-2202, Vector Lab) according to manufacturer instructions. For double staining, first and second primary antibodies were diluted in Dako antibody diluents with 3% goat serum then incubated for 1 h at 37 °C in a humidified chamber. After PBS washes, the sections were covered with secondary antibody (Dako antibody diluents with 3% of goat serum mixed with mouse or rabbit Alexa Fluor® 555 or Alexa Fluor® 660 (Thermofisher)) for 40 min at 37 °C in a humidified chamber. The sections were secured with fluorescent mounting medium containing DAPI and protected with coverslips. Fluorescence was recorded using an Axio Imager M2 imaging microscope (Zeiss, Oberkochen, Germany) and analyzed on digital photographs using Image J software (https://imagej.nih.gov/ij/). Following primary antibodies were used: Rat monoclonal Anti-p21 (ab107099 Abcam 1:50), rabbit anti-CD31 (ab182981, Abcam, 1:500); rabbit anti-MUC1 (ab109185, Abcam, 1:200); rabbit anti-CD34 (ab81289, Abcam, 1:200); mouse anti-CDKN2A/p16INK4a (ab54210, 1:200); mouse anti-PCNA (proliferating cell nuclear antigen) (ab29, Abcam, 1:200); mouse anti-BrdU (ab8152, Abcam, 1:20). BrdU immuno-histostaining was performed by using BrdU immunohistochemistry kit following provider instruction (ab125306, Abcam).

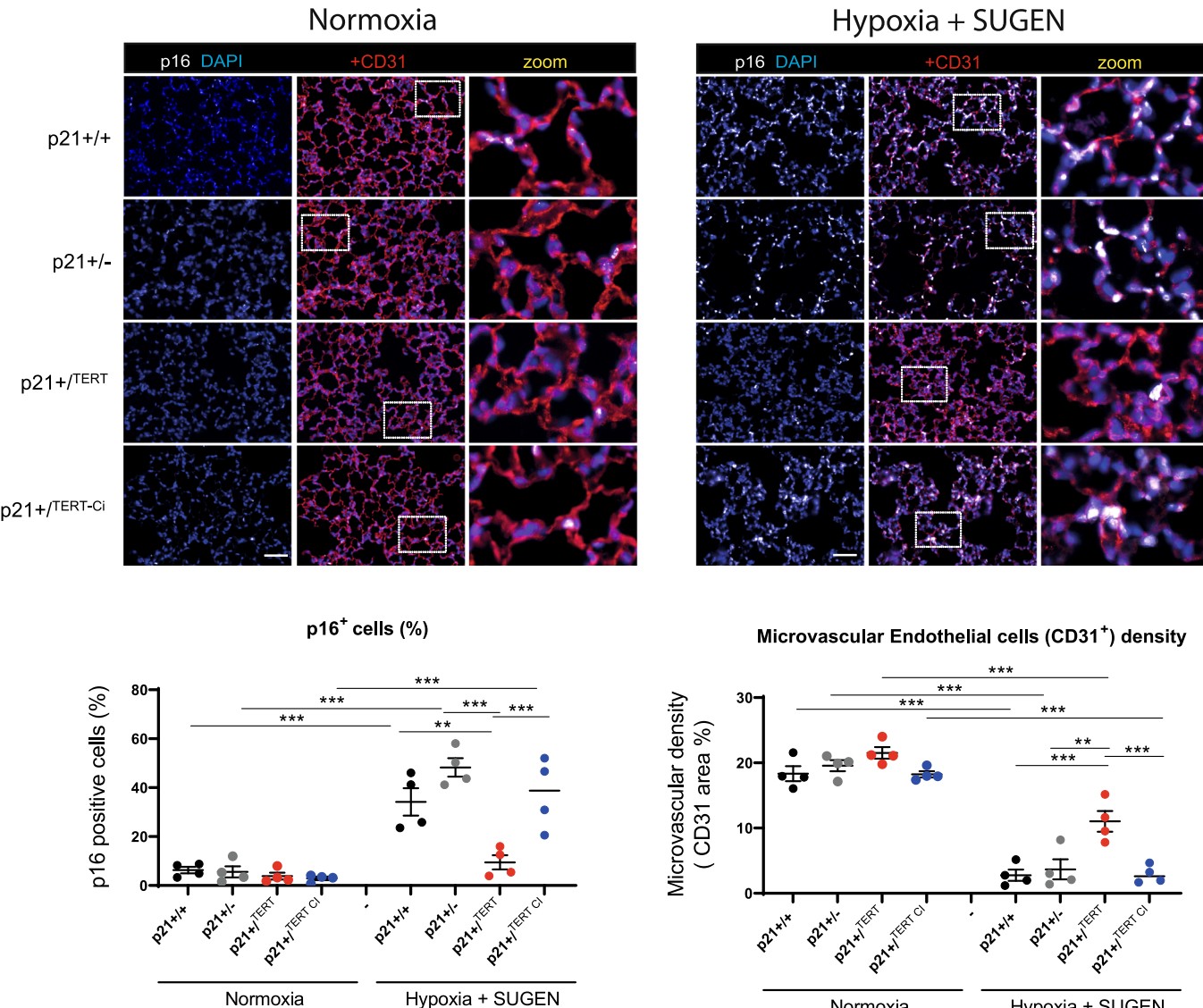

**Figure 10. p21$^{+/Tert}$ mice exposed to SU5416 plus hypoxia are protected against endothelial cell senescence and microvascular rarefication.**

(Upper panel) Representative micrographs showing immunofluorescence of p16 (white, a marker of senescence) and CD31 (red, a marker of endothelial cells) in lung of young mice exposed to normoxia (left) or chronic hypoxia for 21 days (right). Blue—DAPI nuclear staining. Scale bar—50 μm. (Lower panel) Scatter-plot graphs showing cellular senescence (% of p16$^+$ cells, left) and microvascular density (right) in the indicated groups of mice. Data are expressed as individual values per mice and mean ± SEM for group of mice. **$p < 0.01$, ***$p < 0.001$ (one-way ANOVA with Bonferroni post hoc test). Source data are available online for this figure.

## RNA extraction, RT-qPCR, and Western blots

Total RNA was extracted using the method (Chomczynski and Sacchi, 1987) with reagents included in the RNeasy kit (Qiagen). To exclude contamination with genomic DNA, the RNA was treated with DNase I directly on mini columns. Reverse transcription was performed using 0.5–2.0 μg of RNA, 50 ng/μL random hexamers, and 200 U of Superscript IV (Invitrogen) in the 20 μL reaction volume for 15 min at 55 °C, followed by inactivation for 10 min at 80 °C. The resulting cDNA was diluted 5–20-fold and analyzed by real-time qPCR using the SYBR Green master mix (Takara bio) and 400 nanomolar each of the following primers: mTert-2253S 5'-AGCCAAGGCCAAGTCCACAA and mTert-2399A, 5'-AGAGAT

GCTCTGCTCGATGACA to target all *Tert* transcripts/isoforms; 2A-F2 5'-AGCAGGAGATGTTGAAGAAACCC and mTert-5'R25'-GGCCACACCTCCCGGTATC to target *mCherry-2A-Tert* transcript from the KI *Cdkn1a* allele; mActb-90S 5'-ACACCCGC-CACCAGTTCG and mActb-283A 5'-GGCCTCGTCACCCACA-TAGG to target β actin transcript. At least one primer within each pair was designed to anneal onto the exon-exon junction to avoid priming on genomic DNA, and the specificity of amplification was verified on agarose gel.

Immunoblots were carried out using standard procedures. The membranes were incubated with primary antibodies followed by incubation with the secondary HRP conjugates, and the signal was detected using an enhanced chemiluminescence detection system (GE

Healthcare). The unsaturated images were acquired using ChemiDoc MP imaging system (Bio-Rad), and the signal densities were quantified using Image Studio Lite ver 5.2 (LI-COR). Antibodies used were rabbit monoclonal anti-p21 [EPR3993] (abcam) and mouse anti-b-actin (SigmaAldrich A1978) antibody (Table 1).

## Telomere repeat amplification protocol (TRAP)

To measure telomerase activity in cellular/tissue extracts TRAP was performed according to the original protocol (Kim and Wu, 1997; Kim et al, 1994). Briefly, the cells or tissues were extracted on ice using standard CHAPS buffer (60 μL per $10^6$ cells) supplemented with a cocktail of protease inhibitors (Roche) and 20 units of RNase inhibitor (Applied Biosystems) followed by centrifugation at $18,000 \times g$ for 30 min at 4 °C. The protein concentration in the supernatant was quantified using Pierce™ 660 nm Protein Assay and the aliquots equivalent to 800, 400, and 200 ng of protein were used to extend 50 ng of the telomerase substrate (TS oligo) in the 25 μL reaction volume. For gel-based detection of the TRAP product we used FAM-labeled TS and the standard ACX and NT primers (Kim et al, 1994) to amplify the telomerase extension products and the internal TSNT control, respectively. For the real-time qPCR-based TRAP we used unlabeled TS and the ACX primer alone. The qTRAP was performed using the SYBR Green PCR mix, and the amount of CHAPS extracts was optimized by serial dilutions.

## Telomere shortest length assay (TeSLA)

TeSLA was performed according to the protocol described by Lai et al, (2017). Briefly, 50 ng of undigested genomic DNA was ligated with an equimolar mixture (50 pM each) of the six TeSLA-T oligonucleotides containing seven nucleotides of telomeric C-rich repeats at the 3′ end and 22 nucleotides of the unique sequence at the 5' end. After overnight ligation at 35 °C, genomic DNA was digested with CviAII, BfaI, NdeI, and MseI, the restriction enzymes creating either AT or TA overhangs. Digested DNA was then treated with Shrimp Alkaline Phosphatase to remove 5′ phosphate from each DNA fragment to avoid their ligation to each other during the subsequent adapter ligation. Upon heat-inactivation of phosphatase, partially double-stranded AT and TA adapters were added (final concentration 1 μM each) and ligated to the dephosphorylated fragments of genomic DNA at 16 °C overnight. Following ligation of the adapters, genomic DNA was diluted to 20 pg/μL, and 2–4 μL was used in a 25 μL PCR reaction to amplify terminal fragments using primers complementary to the unique sequences at the 5' ends of the TeSLA-T oligonucleotides and the AT/TA adapters. FailSafe polymerase mix (Epicenter) with 1× FailSafe buffer H was used to amplify G-rich telomeric sequences. Entire PCR reactions were then loaded onto the 0.85% agarose gel for separation of the amplified fragments. To visualize telomeric fragments, the DNA was transferred from the gel onto the nylon membrane by Southern blotting procedure and hybridized with the $^{32}$P-labeled (CCCTAA)$_3$ probe. The sizes of the telomeric fragments were quantified using TeSLA Quant software (Lai et al, 2017). Primers are described in Table 1.

## Statistical analysis

Basic statistics, tests for the differences between the means, and one-way analysis of variance (ANOVA) were performed with the GraphPad Prism 9 Software. ANOVA followed by Bonferroni multiple comparison test was used to compare the means of more than two independent groups. Pairwise correlation and multiple regression analyses were performed using Statistica 13.0 software package (StatSoft/Dell).

## Single-cell RNAseq

### Isolation of lung single cells from 18-month-old p21⁺/⁺ (1×), p21⁺/⁻ (3×), p21⁺/Tert (5×), and p21⁺/Tert (2×) mice

Mouse trachea is injected with 1.5 mL dispase 50 U/mL (Corning #354235) followed by 0.5 ml agarose 1%. Lung is resected and minced in 3 ml DPBS 1× CaCl²⁺ and MgCl²⁺ (Gibco, 14040-091). Collagenase/dispase 100 mg/mL (Roche #11097113001) is added to the chopped tissue and placed on a rotator at 37 °C for 30 min. Enzymatic activity is inhibited by adding 5 mL of DPBS 1× (Gibco, 14190-094) containing 10% FBS and 1 mM EDTA (Sigma #E7889). Cell suspension is filtered through 100 μm nylon cell strainer (Fisher Scientific #11893402), treated by DNase I (Sigma D4527) and filtered again through a 40 μm nylon cell strainer (Fisher Scientific #11873402). Red blood cells are removed by red blood cell lysis (Invitrogen, 00-4333-57) treatment for 90 s at room temperature. Finally, lung cells are centrifuged at $150 \times g$, 4 °C for 6 min, resuspended in DPBS 1× containing 0.02% BSA (Pan Biotech #P06-13911000) and counted in a Malassez.

### Preparation of single-cell sequencing libraries

Single-cell 3'-RNA-Seq samples were prepared using single cell V2 reagent kit and loaded in the Chromium controller according to standard manufacturer protocol (10× Genomics, PN-120237) to capture 6000 cells. Briefly, dissociated lung cells are encapsulated using microfluidic device. RNAs are captured on beads coated of oligos containing an oligo-dTTT, UMIs and a specific barcode. After reverse transcription, cDNAs are washed, PCR-amplified and washed again before analysis on a Bioanalyzer (Agilent) for quality control. Finally, libraries are prepared following standard Illumina protocol and sequenced on a NovaSeq sequencer (Illumina). Raw sequences are demultiplexed and reads are mapped onto the mm10 reference genome using the v2.3 Cell Ranger pipeline (10× Genomics) to generate a count matrix for each sample.

### Quality control and normalization of the single-cell data

The digital matrices were filtered by cell type (on clusters composed of the same cell type), to remove low-quality cells with low UMI counts and cells with relatively high mitochondrial DNA content. Outlier analysis was performed with perCellQCMetrics from the scatter package. An upper cutoff was manually determined for each sample based on a plot of gene count versus UMI count or % of mitochondrial genes, to have at least 1000 UMIs, number of transcripts ranging between 1000 and 30,000 and at most 14% mitochondrial transcripts. The quality was consistent across samples, and differences in RNA and gene content could be ascribed to cell-type-specific effects. DGE matrices from all samples, sequenced at different time, were then merged and subsequently normalized using the deconvolution normalization method in the scran R package in order to correct for differences in read depth and library size inside and between samples.

### Clustering, cell type, and cycle annotation

Seurat v4.1.0 was used to perform dimensionality reduction, clustering, and visualization on the unique combined (with all

samples) and normalized matrix. After scaling the data, dimensionality reduction was performed using PCA on the highly variable genes. Seurat's *FindNeighbors* function was run to identify cluster markers with the following parameters: reduction = "pca" and dims = 1:10, followed by the *FindClusters* function with resolution = 1. The Seurat function *RunUMAP* was used to generate two-dimensional umap projections using the top principal components detected in the dataset. To annotate cells, we mapped each cluster to the Mouse Cell Atlas (Kalucka et al, 2020) by using the *scMCA* function (Sun et al, 2019). Cluster identities were further verified according to gene markers found with the *FindMarkers* function from Seurat. Doublet cells were identified manually as expressing markers for different cell types and were removed from the matrix. In order to define endothelial (EC) subclusters, EC cell classes were isolated and subjected to a new clustering by the *FindNeighbors* function with new parameters: reduction = "pca" and dims (1:15 for EC). Seurat *FindAllMarkers* function was applied to each EC subclusters and the markers were compared to the top 50 gene markers previously found in lung EC (Gillich et al, 2020). We thus annotated EC subclusters depending on the enrichment of marker genes and identified five subtypes of EC (artery, capillary_1, capillary_2, vein and lymphatic). We annotated two functionally divergent lung EC capillary clusters according to Gillich et al, 2020. We thus reassign EC capillary_1 as "aCap" and EC capillary_2 as "gCap." In order to assign cell cycle phase to each cell, we used the Cyclone method to our single-cell RNA-Seq dataset.

## Data availability

Original data for single-cell RNA-sequencing are available at the Gene Expression Omnibus (GEO; https://www.ncbi.nlm.nih.gov/geo/query/acc.cgi, GEO accession: GSE165218). TIF (Telomere induced foci) images are stored in OMERO server at the French "Service Numerique de Bioimagerie." No permanent link can be generated to have access to the images. Dedicated link will be generated upon request.

## Peer review information

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

## Acknowledgements

We thank Lea Harrington for providing m*Tert*-/- mouse ES cells. We are very grateful to Myriam Grunewald (Hebrew University of Jerusalem) for critically reading the manuscript. We are grateful to the CRCM animal facility for taking care of the mouse strain colonies, to Manon Richaud from the CRCM flow cytometry and cell-sorting platform, and to Lionel Spinelli and Arnaud Guille for useful discussions and technical support for the single-cell analysis. We thank the Institut Curie Bioinformatics platform for data management and quality control of single-cell data and the CRCM DNA Damage Platform and Imaging facility. Work in VG's Lab is supported by "La Ligue Contre le Cancer", Equipe Labellisée, The Canceropole PACA (Projet Emergent), the "Institut National du Cancer" (INCA), PLBIO 2019, and the cross-cutting Inserm program on aging (AGEMED). SA's Lab is supported by grants from the Inserm, Ministère de la Recherche, Agence Nationale pour la Recherche (ANR), Institut National du Cancer (INCA), and Fondation pour la Recherche Médicale (FRM), et la Fondation ARC. AL-V's lab is supported by INCa (PLBIO2019) and La Ligue contre le cancer-Paris (RS21/75-24). SC-A is a recipient of a European CO-FUND PhD fellowship from Institut Curie (European Union's Horizon 2020 research and innovation program under the Marie Skłodowska-Curie grant agreement No 666003). IF's lab was funded by grants from the Spanish Ministry of Science and Innovation (PID2019-110339RB-I00) and the Comunidad de Madrid (S2017/BMD-3875). EG's lab was supported by the Fondation ARC (Program ARC), and the cross-cutting Inserm program on aging (AGEMED). CIPHE is supported by PHENOMIN (French National Infrastructure for mouse Phenogenomics; ANR10-INBS-07).

## Author contributions

**Larissa Lipskaia**: Conceptualization; Formal analysis; Validation; Investigation; Visualization; Methodology; Writing—review and editing. **Marielle Bréau**: Conceptualization; Validation; Investigation; Visualization; Methodology. **Christelle Cayrou**: Conceptualization; Validation; Investigation; Methodology. **Dmitri Churikov**: Conceptualization; Formal analysis; Supervision; Validation; Investigation; Visualization; Methodology; Writing—review and editing. **Laura Braud**: Investigation; Methodology. **Juliette Jacquet**: Investigation; Methodology. **Emmanuelle Born**: Investigation; Methodology. **Charles Fouillade**: Investigation; Methodology. **Sandra Curras-Alonso**: Investigation; Methodology. **Serge Bauwens**: Investigation; Methodology. **Frederic Jourquin**: Project administration. **Frederic Fiore**: Conceptualization; Validation; Investigation; Methodology. **Rémy Castellano**: Investigation; Methodology. **Emmanuelle Josselin**: Methodology. **Carlota Sánchez-Ferrer**: Investigation; Methodology. **Giovanna Giovinazzo**: Investigation. **Christophe Lachaud**: Conceptualization; Validation; Investigation; Methodology. **Eric Gilson**: Conceptualization; Funding acquisition; Investigation. **Ignacio Flores**: Conceptualization; Investigation; Methodology. **Arturo Londono-Vallejo**: Conceptualization; Formal analysis; Supervision; Investigation; Methodology; Writing—review and editing. **Serge Adnot**: Conceptualization; Formal analysis; Supervision; Funding acquisition; Validation; Investigation; Visualization; Methodology; Writing—review and editing. **Vincent Géli**: Conceptualization; Formal analysis; Supervision; Funding acquisition; Validation; Investigation; Visualization; Writing—original draft; Project administration.

## Funding

## Disclosure and competing interests statement

The authors declare no competing interests.

# Expanded View Figures

**Figure EV1.  Validation of p21$^{+/Tert}$ model.**

(**A**) p21$^{+/+}$ and p21$^{+/Tert}$ littermates were subjected to whole-body ionizing radiation (1.5 gray). The fluorescence emitted by the mCherry was followed post-irradiation at the indicated times by in vivo mCherry imaging (Excitation = 545 nm, Background = 495 nm, Emission = 615 nm). (**B**) Left panel, p21 expression and mCherry fluorescence were analyzed in the liver and kidneys after doxorubicin treatments. Right panel, livers and kidneys of p21$^{+/+}$ and p21$^{+/Tert}$ mice were harvested 24 h after doxorubicin treatment. The level of p21 protein was evaluated by semi-quantitative immunoblotting and liver and kidney were imaged for mCherry fluorescence. (**C**) Primers to distinguish transgene expression from total Tert expression are shown. (**D**) *Tert* RT-qPCR experiments were performed using RNA extracted from the harvested organs. The mean and the SEM of the three biological PCR replicates are plotted. For both organs, the difference between the doxorubicin-treated and untreated samples is statistically significant ($p < 0.001$, unpaired Student's *t* test) for the p21$^{+/Tert}$ but not p21$^{+/+}$ mice.

A

Whole-body IR (1 gray)

p21⁺/⁺
p21⁺/Tert → follow mCherry fluorscence

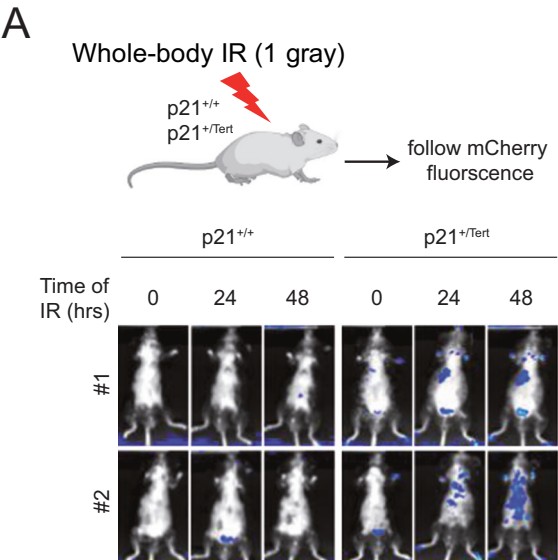

B

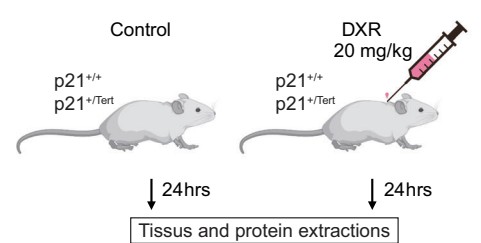

| Organ | Liver | | | | Kidney | | | | |
|---|---|---|---|---|---|---|---|---|---|
| Genotype | p21⁺/⁺ | | p21⁺/Tert | | p21⁺/⁺ | | p21⁺/Tert | | |
| Doxorubicin | - | + | - | + | - | + | - | + | |

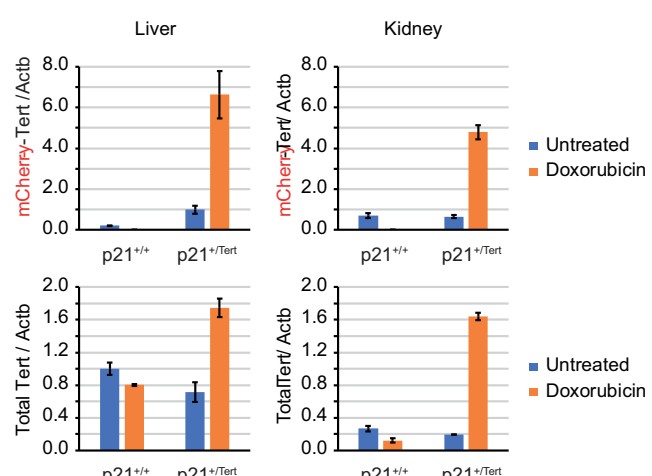

C

Primers for the analysis of total and mCherry-mTert mRNA levels

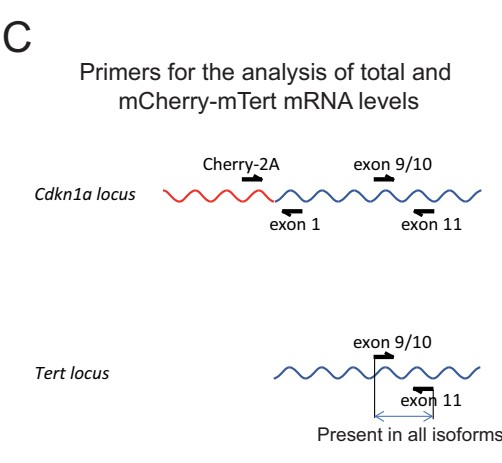

D

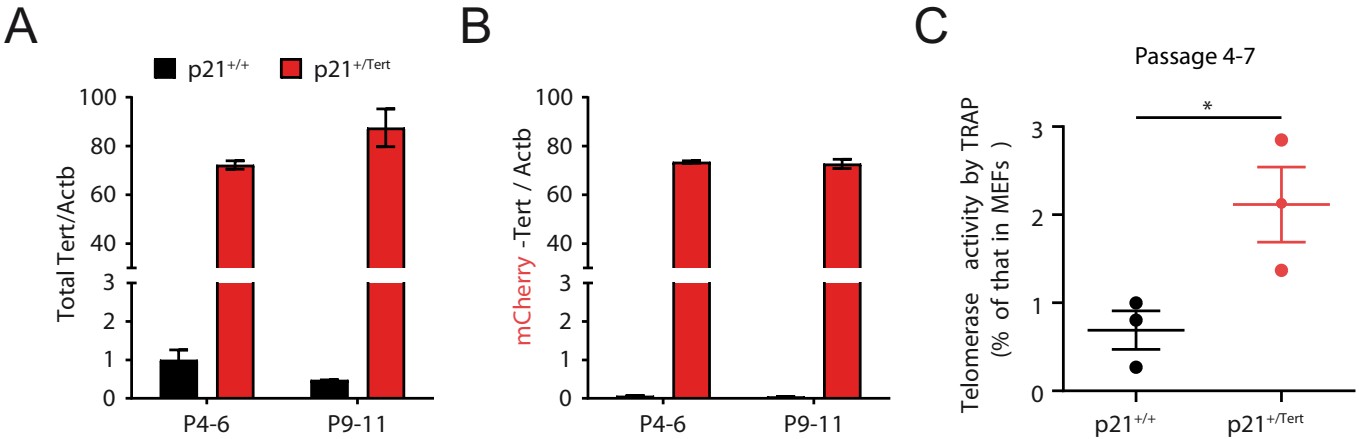

**Figure EV2.   p21-promoter dependent mTert expression bypasses senescence in PA-SMCs ex vivo (related to Fig. 1C).**

(A, B) Quantification of the m*Tert* mRNA levels in PA-SMCs from the p21$^{+/+}$ and p21$^{+/Tert}$ 4-month-old mice (littermates) at early (p4-6) and late (p9-11) passage. Left panel represents the level of total *Tert* mRNA (transcribed from both the native Tert locus and KI allele), while right panel represents the level of *Tert* mRNA transcribed from KI allele only. Nearly all *Tert* mRNA is transcribed from the KI allele. The means of three independent measurements are plotted, and the error bars are SEs. The difference in the level of *Tert* mRNA between the p21$^{+/+}$ and p21$^{+/Tert}$ cells is highly significant ($p < 0.001$, unpaired Student's *t* test) for both early and late passages. (C) Telomerase activity measured by qTRAP at early passages. The data points correspond to vascular PA-SMC cultures established from individual p21$^{+/+}$ and p21$^{+/Tert}$ 4-month-old mice. The data are expressed as the mean ± SEM. *$p < 0.05$ from the two-sided *t* test. (D, E) Analysis of the short telomere fraction by Telomere Shortest Length Assay (TeSLA) in the cultured PA-SMCs from p21$^{+/+}$ and p21$^{+/Tert}$ mice. Southern blots probed for the TTAGGG repeats in (A) and quantification of the cumulative number of short telomeres across the telomere length thresholds in (E). Source data are available online for this figure.

A

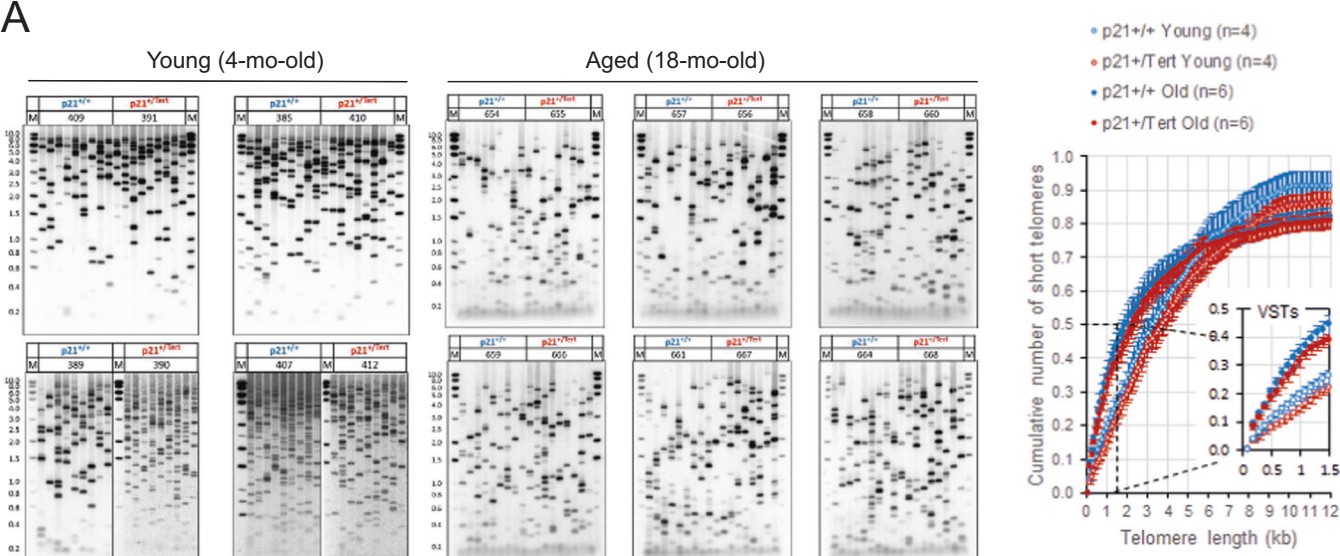

B

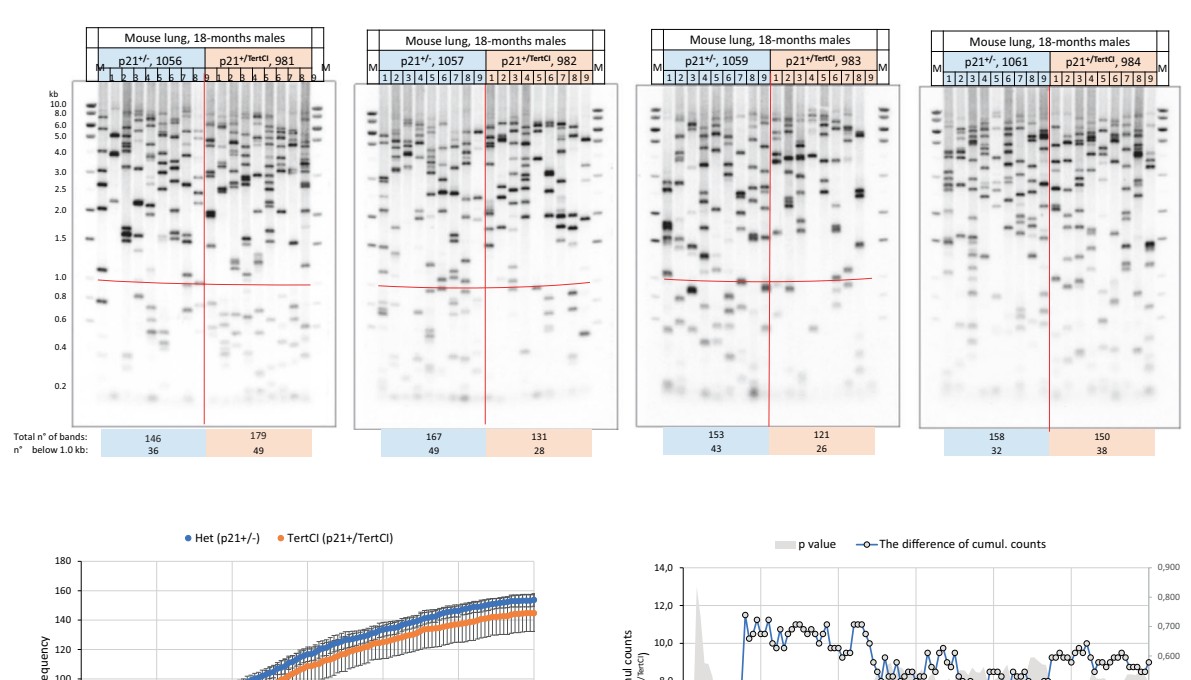

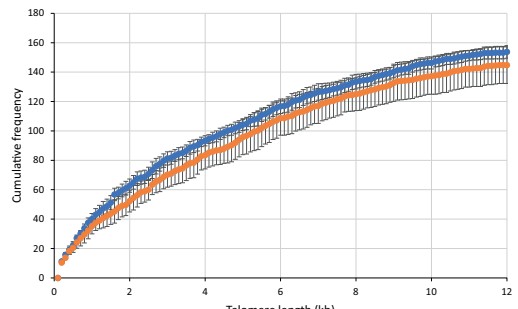

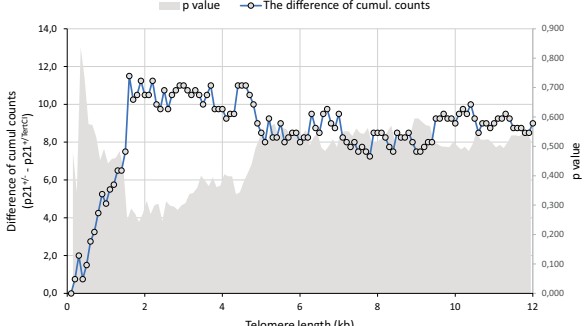

Figure EV3. Analysis of the individual short telomeres by TeSLA in the lungs of the young and old mice.

(A) TeSLA Southern blots depicting short telomeres in the lungs of the p21$^{+/+}$ and p21$^{+/Tert}$ littermates (4 and 18-month-old mice). Genomic DNA was extracted from whole lungs and the length of individual short telomeres was determined by TeSLA. Note that young mice, regardless of their genotype, have less telomeres shorter than 1 kb compared to the old mice. The graph depicts cumulative number of short telomeres per genome in the range of 0.2–12 kb. The area corresponding to the very short telomeres (VSTs) is magnified in the inset. The mean values ± SEM are plotted. (B) TeSLA Southern blots depicting short telomeres in the lungs of the p21$^{+/-}$ and p21$^{+/TertCl}$ mice (18-month-old). TeSLA Southern blots are shown on top. The graph on the bottom left shows the cumulative frequency of short telomeres in the range of 0.2–12 kb for the mice of two genotypes. The graph on the bottom right depicts the difference of cumulative counts for 0.1 kb bins between the two genotypes (left $y$ axis) and the corresponding $p$ values from the two-tailed $t$ test (right $y$ axis). The difference is not significant for any bin indicating that Tert$^{Cl}$ is unable to improve the load of the short telomeres in old mice.

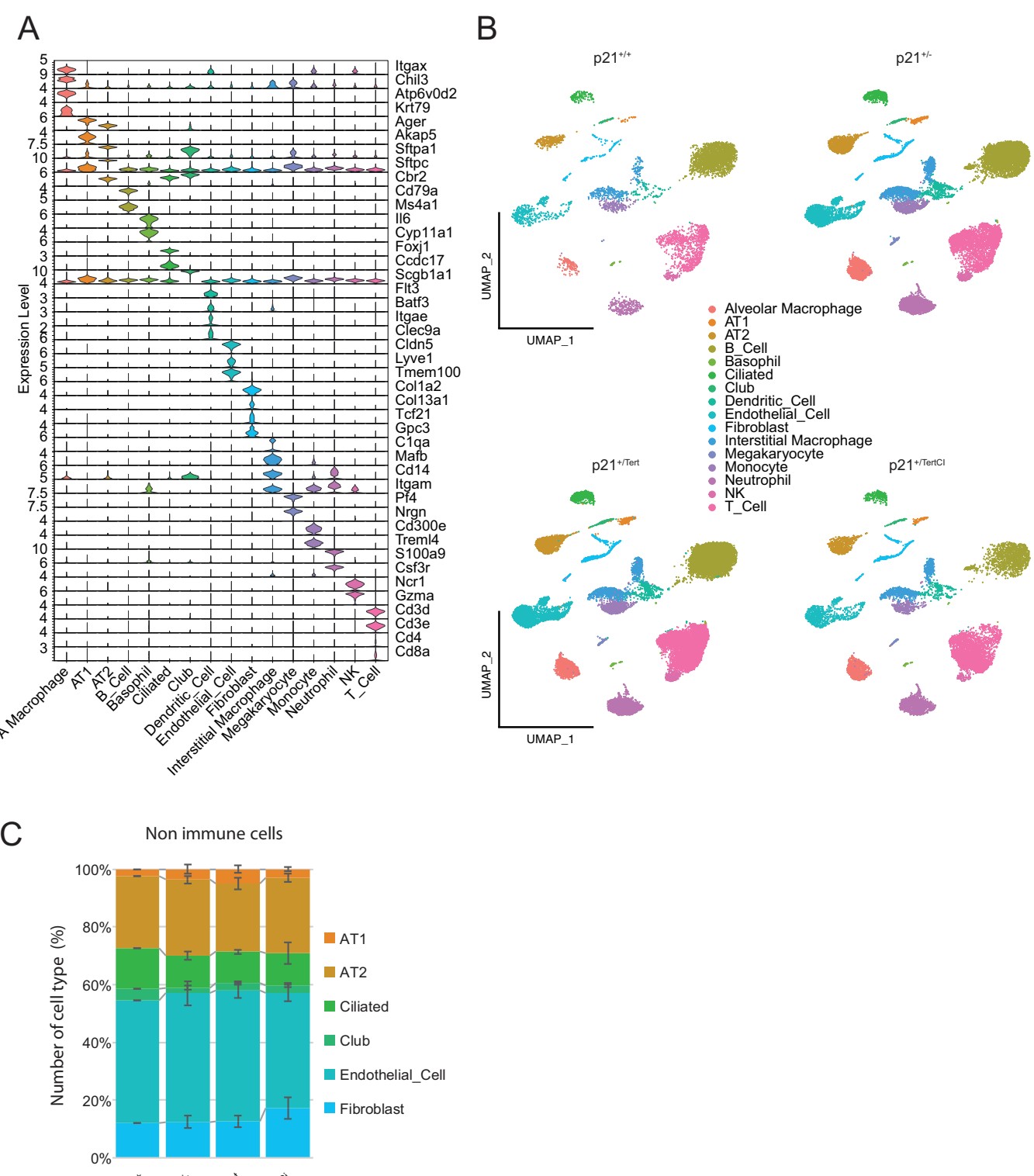

**Figure EV4. Lung cell types in the four mouse models.**

(A) Representative markers use to annotate lung cell types in the four mouse models. (B) UMAP clustering of lung cells. Lung cell populations were identified in lung samples from WT (p21$^{+/+}$), p21$^{+/-}$, p21$^{+/Tert}$ and p21$^{+/TertCl}$ 18-month old mice. (C) Quantification of the cell types in lungs of the mice of the 4 indicated genotypes. At least 3 mice of each genotype were analyzed. The mean values ± SEM are plotted.

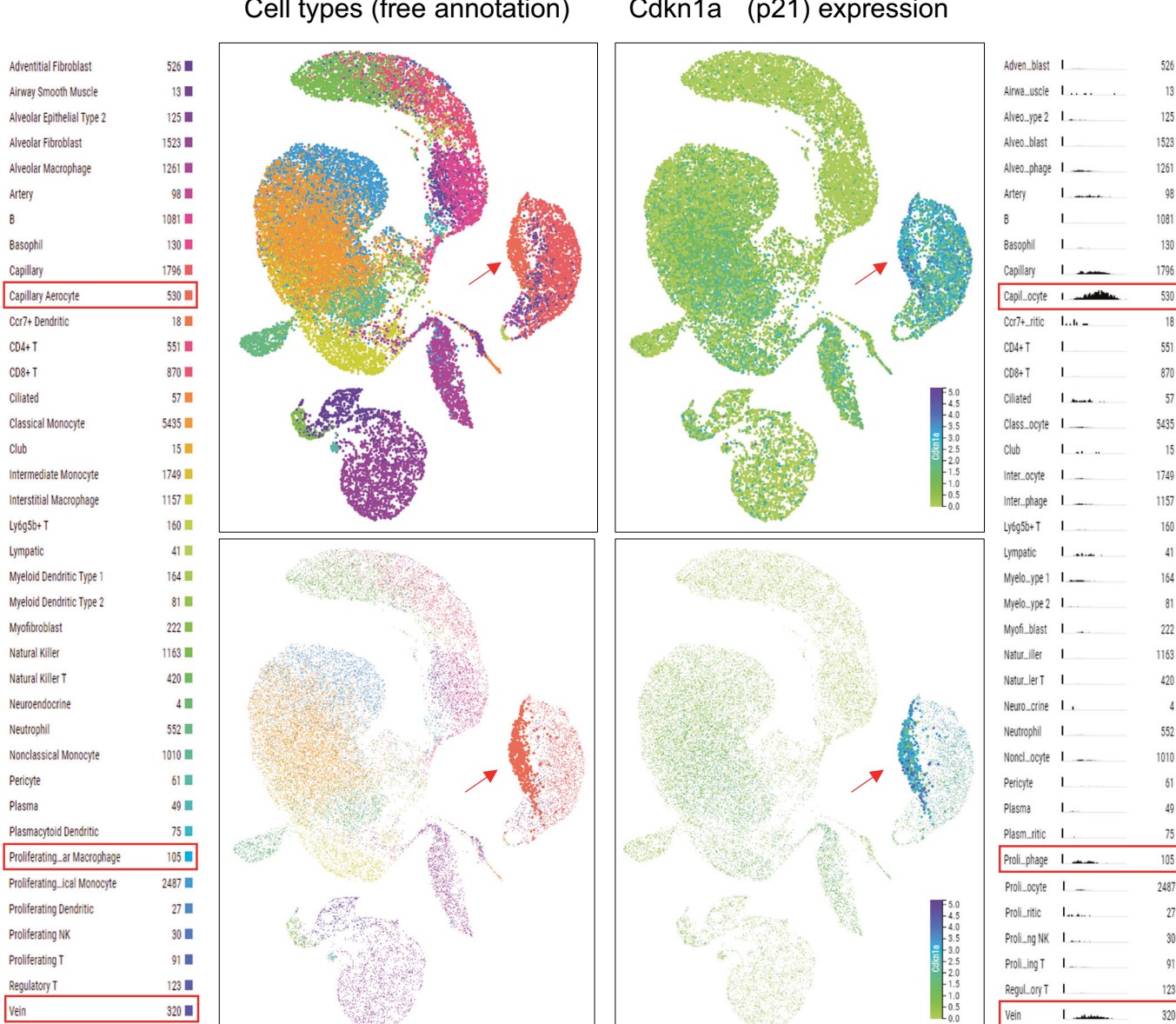

**Figure EV5.  UMAP plots generated using lung droplet scRNA-seq data in Tabula Muris Senis.**

The data for mice of all ages (1–30 months) were included. In the top panels all cell types are shown, while in the bottom panels only the capillary aerocyte population is highlighted. Red arrow points to the capillary aerocytes, and red boxes in the annotation mark cell types with elevated p21 expression level. Histograms (on the right) show scaled number of cells expressing p21 (y axis) versus expression level (x axis).

