## [Peer Review File · EMBO Reports]

mTert induction in p21-positive cells counteracts capillary rarefaction and pulmonary emphysema

Larissa Lipskaia, Marielle Bréau, Christelle Cayrou, Dmitri Churikov, Laura Braud, Juliette Jacquet, Emmanuelle BORN, Charles Fouillade, Sandra Curras-Alonso, Serge Bauwens, Frederic Jourquin, Frederic FIORE, Rémy Castellano, Emmanuelle Josselin, Carlota Sánchez-Ferrer, Giovanna Giovinazo, Christophe Lachaud, Eric Gilson, Ignacio Flores, Arturo Londono-Vallejo, Serge Adnot, and Vincent Geli

DOI: [10.15252/embr.202358279](https://doi.org/10.15252/embr.202358279)

Corresponding author(s): Vincent Geli (vincentgeli1@gmail.com) , Serge Adnot (serge.adnot@inserm.fr)

Review Timeline:

Transfer Date:	6th Oct 23
Editorial Decision:	9th Oct 23
Revision Received:	16th Nov 23
Editorial Decision:	5th Dec 23
Revision Received:	12th Dec 23
Accepted:	14th Dec 23

Editor: Achim Breiling

Transaction Report: This manuscript was transferred to EMBO reports following peer review at The EMBO Journal.

Dear Dr. Geli,

Thank you for transferring your revised manuscript to EMBO reports. I now went through the manuscript, the referee reports from The EMBO Journal (attached again below) and your revision plan. The referees have several remaining concerns and suggestions to improve the manuscript, or to strengthen the data and the conclusions drawn.

EMBO reports emphasizes novel functional over detailed mechanistic insight, but asks for strong in vivo relevance of the findings, and clear experimental support of the major conclusions. Thus, we will not require addressing points regarding more mechanism experimentally. However, it will be necessary that during final revision you address all points questioning the main conclusions of the study, and all technical concerns, or points regarding the experimental designs, model systems used, or data presentation.

Given the constructive referee comments, I would like to invite you to revise your manuscript with the understanding that all concerns of the referees must be addressed in the revised manuscript or in a detailed point-by-point response as indicated in your revision plan.

It will be important that in the re-revised study you include the data mentioned in your response letter (revision plan), and that you re-write the study as asked by the referees (making it clear "... that this is a lung p21 paper and not a telomere length paper").

- Please provide the abstract written in present tense throughout.
- Please provide a new completed author checklist (for EMBO reports), which you can download from our author guidelines (<https://www.embopress.org/page/journal/14693178/authorguide>). Please insert page numbers in the checklist to indicate where the requested information can be found in the manuscript. The completed author checklist will also be part of the RPF.

Please also follow our guidelines for the use of living organisms, and the respective reporting guidelines:
<http://www.embopress.org/page/journal/14693178/authorguide#livingorganisms>

- Regarding data quantification and statistics, please make sure that in the final manuscript file the number "n" for how many independent experiments were performed, their nature (biological versus technical replicates), the bars and error bars (e.g. SEM, SD) and the test used to calculate p-values is indicated in the respective figure legends (also for main and EV figures). Please also check that all the p-values are explained in the legend, and that these fit to those shown in the figure. Please provide statistical testing where applicable. Please avoid the phrase 'independent experiment', but clearly state if these were biological or technical replicates. Please also indicate (e.g. with n.s.) if testing was performed, but the differences are not significant. In case n=2, please show the data as separate datapoints without error bars and statistics.

See also:

<http://www.embopress.org/page/journal/14693178/authorguide#statisticalanalysis>

- Please add scale bars of similar style and thickness to all microscopic images, using clearly visible black or white bars (depending on the background). Please place these in the lower right corner of the images themselves. Please do not write on or near the bars in the image but define the size in the respective figure legend.

- Please order the manuscript sections like this using these names:

Title page - Abstract - Keywords - Introduction - Results - Discussion - Materials and Methods - Data availability section (DAS) - Acknowledgements - Disclosure and Competing Interests Statement - References - Figure legends - Expanded View Figure legends

- In the Data Availability section, please make sure to include a direct URL allowing to access the referenced datasets (via their accession number) from the relevant databases (such as GEO). Please provide accession ID provided for "TIF Images are stored in OMERO server at the French "Service Numerique de Bioimagerie"." Please also ensure that datasets will become rapidly released to the public at this point.

- Please note our reference format (we need up to 10 authors listed before 'et al'). See:
<http://www.embopress.org/page/journal/14693178/authorguide#referencesformat>

- We updated our journal's competing interests policy in January 2022 and request authors to consider both actual and

perceived competing interests. Please review the policy <https://www.embopress.org/competing-interests> and add a statement declaring your competing interests. Please name that section 'Disclosure and Competing Interests Statement' and add it after the author contributions section.

- Please make sure that all the funding information is also entered into the online submission system and is complete and similar to the one in the manuscript text file (in the Acknowledgements). It seems that presently RS21/75-24, No 666003, S2017/BMD-3875, ANR-10-EQPX-03 (Equipex) and ANR-10-INBS-09-08 (France Génomique Consortium), SiRIC Grant INCa-DGOS-465 and INCa-DGOS-Inserm_12554 are not entered into the submission system. Please check.

- Please include Appendix Fig. S1 in a proper Appendix file. This needs to be a single pdf file labeled Appendix. The Appendix should have page numbers and needs to include a table of contents on the first page (with page numbers) and legends for all content.

- There is a "supplementary dataset 1" called out (page 17). What file does this refer to? Please check and update the callout.

- Table EV1 is a dataset. Please name and upload this as 'Dataset EV1' and change the respective callouts.

- Thanks for providing the source data (SD). Presently SD for 2CDEF, 4ACD, 6DEF and 7C appears to be missing. Please check and also upload the completed source data checklist (attached again) with your revised manuscript files. Finally, please upload the SD as one ZIP folder per figure, and one folder for all the EV figure source data.

- Please upload the information provided as "SUPPLEMENTAL INFORMATIO" as reagents and tools table. I have attached templates for that in word or excel format. Please upload the filled in table to the manuscript tracking system as 'Reagent Table' file. Please also adjust any callouts to this table. The example linked below shows how the table will display in the published article and includes examples of the type of information that should be provided for the different categories of reagents and tools. Please list your reagents/tools using the categories provided in the template and do not add additional subheadings to the table. Reagents/tools that do not fit in any of the specific categories can be listed under "Other":
https://www.embopress.org/pb%2Dassets/embo-site/msb_177951_sample_FINAL.pdf

In addition, I would need from you:

- a short, two-sentence summary of the manuscript (not more than 35 words).

- two to four short (!) bullet points highlighting the key findings of your study (two lines each).

- a schematic summary figure as separate file that provides a sketch of the major findings (not a data image) in jpeg or tiff format (with the exact width of 550 pixels and a height of not more than 400 pixels) that can be used as a visual synopsis on our website.

I look forward to seeing a revised version of your manuscript when it is ready. Please let me know if you have questions or comments regarding the revision.

Kind regards,

Achim

Referee #1:

This referee would like to commend the authors on the substantial work done to address most of the minor and major concerns raised from the original manuscript. Especially the addition of the hypoxia model of stress-induced emphysema (Figure 9/10) demonstrating the rescued phenotype of p21+/Tert mice, validates further their main observations.

However, there are still outstanding points that have not been addressed sufficiently. More importantly, however, some of the new data raises further concerns that their model is not functioning as hypothesized. It is already known that telomere dysfunction plays a crucial role in lung aging and COPD and the authors acknowledge it by citing recent reviews and research work (Birch et al 2015, Rossiello et al. 2022), but their new TIF/p21 data suggest otherwise. They "assume that induction of the mTert expression from activated p21 promoter might repair telomeres" but their data do not support this hypothesis. This raises potential issues or technical limitations of their model. More specifically, the study reveals that p21 promoter driven TERT protects against development of ultrashort telomeres, but not against development of dysfunctional telomeres (or from p21

upregulation). However, ultrashort telomeres do not activate cellular senescence, dysfunctional telomeres do. Therefore, unless the authors have evidence that ultrashort telomeres cause cellular senescence or cellular dysfunction in the absence of telomere dysfunction, the beneficial effects of TERT expressed from the p21 promoter (ie. reduced senescence, a decline in capillary density and protection from lung emphysema development) must be due to a function of TERT that does not involve its ability to suppress cellular senescence due to telomere dysfunction. Their conclusion that "TERT delays senescence ... by counteracting telomere shortening..." therefore is likely incorrect.

Furthermore, given the new data demonstrating that p21^{+/Tert} mice develop fewer p16 and SA-bGal positive cells, it seems that p21^{+/Tert} suppresses cellular senescence in a manner that is dependent on functional telomerase activity but independent of its ability to protect cells from developing dysfunctional telomeres. These data are difficult reconcile with our current understanding of telomerase function.

It is understandable that several controls missing from some panels were due to unprecedented circumstances during the pandemic, but the absence of key controls makes the interpretation of the in vivo data partially inconclusive.

Major outstanding concerns summarized below:

1. The relationship between levels of p21 expression, levels of the p21-mCherry-Tert transgene and telomerase activity remain still unexplored. In Figure 1D levels of mCherry-Tert are shown to be ~10 fold higher in 3-month old p21^{+/Tert} mice compared to p21^{+/+} mice although their respective Cdkn1a levels are shown to be similar. This means that the mTert might not be entirely induced in a p21-dependent manner. This is a main point not addressed in the rebuttal.
2. As mentioned in the previous review, the p21/Tert construct used in the mouse model needs to be evaluated more carefully in cell cultures and/or in the mouse. The pCAG-controlled mCherry-mTert construct used for Fig 1B is not appropriate for evaluating enzymatic activity as it differs from the construct used in the mouse. The key control is whether the p21-induced Tert transgene in the mouse develops enzymatic activity upon p21 promoter activation. These data, using PA-SMC cells, should be shown in the main Figure. Proper controls are cells from all genotypes including from cells expressing the wt and mutant Tert. P21 promoter should be activated due to replicative aging as well as due to DNA damaging drugs. Tert activity and mRNA expression levels should be measured in all cases, especially in light of the observations that Tert activity and mRNA expression seem to be uncoupled (see Figure EV2 in Panels A-C where the authors show ~80-fold induction of the transgene mRNA in the cultured PA-SMCs, but only a 2-fold increase in telomerase activity. Why is this?). These are not trivial questions, since all the conclusions of the in vivo data regarding the mechanism of how p21 induced mTert alleviates emphysema in the old lung are based on the premise that the transgene is expressed and translated properly, that the transgene has enzymatic activity, and that enzymatic activity is increased upon DNA damage as well as in response to telomere dysfunction.
3. The Kaplan-Meier survival curves included in rebuttal show no statistical differences between groups, but do show clear trends of increased mortality in the mice p21^{+/-}, p21^{+/Tert} (50% survival rate). This needs to be discussed in the text, since Tert expressed from a p21 promoter (or p21 haploinsufficiency) does not appear to benefit the organism as a whole.
4. If p21 protein levels are perturbed in all genotypes, and more so in both mTert, mTertCl, as shown in the new Figure 2A it demonstrates that p21 mediated senescence is perturbed (Figure 5E,F SABgal trends lower in p21^{+/TertCl} compared to p21^{+/-}). If "the p21^{+/TertCl} mice exhibit the same aging characteristics in the lung as control mice" it suggests that DNA damage-induced p21-mediated senescence is not a main contributor in age-related lung emphysema, which contradicts the main idea revolved around the generation of the transgene. The authors should analyze abundance of total DDR foci, regardless whether they are telomeric or not, as the data can be easily extracted from their TAF/TIF analysis. A decrease of DDR foci in p21/TERT animals would suggest that TERT exerts its beneficial effects due to its non canonical function in repairing non-telomeric DNA damage.
5. In EV3 graph comparisons of VSTs of p21^{+/+} vs p21^{+/-} and p21^{+/TertCl} must also be shown to understand at what levels haploinsufficiency alone affects the distribution of VSTs (the data are there).
6. The authors should speculate less about the non-telomeric roles of mTert in regulating p21 and CD34, since it's also possible that the mTERT-p21 mRNA hybrid produced from the cassette has additional functions that are confounding to the function of an inducible native mTert - even a possible dominant negative form that affects normal levels of p21.

Although, the addition of the hypoxia mouse model in the revised manuscript provides solid evidence that indeed mTERT can rescue induced emphysema (the main scope of the study), their aging model presents several major limitations (as discussed above) that need to be addressed.

Referee #2:

Comments to revision by reviewer 2:

The authors provided more explanations and arguments without yet did not provide the additional experiments that I asked for. Below I try to clarify my points and requests.

In summary, I am still concerned that the mild effect may be pleiotropic, as clear evidence for telomere length or telomere dysfunction drives the effects is missing. That said, I appreciate the significant amount of work that went into this manuscript. I agree with the authors that the chronic hypoxia data supports their observations, yet again the experiment does not provide

insight into the mechanism(s) at play.

ORIGINAL COMMENT:

-The telomere length analysis is difficult to interpret. It is of course correct that the short 11 telomeres are most relevant but the assay used is strongly biased towards short telomeres. The TESLA assay in the experiment analyzes the telomeres of ~ 80pg of genomic DNA ~ 12 cells per reaction with ~100 telomeres/ cell. Each lane shows about >12 telomeres. This suggests that ~1% of the telomeres in these mice are short. This is unexpected.

AUTHOR RESPONSE:

In our hands, TeSLA amplifies telomeres up to ~15 kb. Assuming a bell-shaped distribution (could be skewed) of mouse telomere lengths with an average around 45-50 kb, only a small proportion of telomeres (~5-10% in one tail of the distribution) can be analyzed by TeSLA. Therefore, if 1% of the telomeres amenable to TeSLA analysis are short, it corresponds to only 0.05-0.1% of total mouse telomeres. The validity of TeSLA for the analysis of short telomeres in mice was addressed in the original publication (Lai et al., Nat Comm 2017).

NEW COMMENT by reviewer 2:

I do not agree with the authors on this point. The authors add approximately 1200- 2400 (after S-phase) telomeres in a tube and then detect 12-20 short telomeres? Where do all these telomeres come from? If the authors DNA quantification is equal and accurate, their data clearly puts the Lai et al. publication into question. Stats and analysis need to be provided, as well as the base assumptions about the telomere length (distribution) above 15kb and how this was included in the stats. These experiments need independent corroboration.

ORIGINAL COMMENT:

The authors need to evaluate this using more standard techniques that have less of a telomere length bias (FISH, pulsed field TRF, etc.).

Yes, there is a small yet appreciable difference in the telomere length in the TERT-p21 het cells. It seems important that the authors establish the constitutional telomere length is the same in all mouse strains. This is to control for the additional copy of TERT elongates -driven by p21- did not increase telomere length in the germline and to control for founder effects during the generation of the mice. In other words, do the authors have any evidence that the p21 regulation matters for the effects seen? Or is the basal p21 expression (-driving TERT) sufficient to elongate telomeres?

AUTHOR RESPONSE:

If we understand well, the reviewer is concerned that telomeres in p21+/Tert mice could be preelongated during either gameto- or embryogenesis due to basal p21 promoter activity. We are not sure that this concern is really relevant, because pre-elongation of telomeres does not protect them from breakage that generates VSTs.

NEW COMMENT by reviewer 2:

I am not sure why the authors say that 'pre-elongation of telomeres does not protect them from breakage that generates VSTs'. Have the authors shown that leads to "the short telomeres" in their model? And even if it was the mechanism, longer telomeres will break into larger residual telomeres don't they? I am not sure why the authors argue against a trivial experiment that takes 3-4 days to perform to check for this.

My request is also motivated by the apparent artifacts revealed by the authors TESLA analysis. Thus, please provide telomere length data using non-TESLA data for somatic tissue ear, tail and lung or anything else that demonstrated that "starting telomere length (see explanation above) that telomere length in these mice is not altered in young mice across genotypes. I suggest to take DNA from 4 mice (young age matched) of each genotype (4) harvest DNA of the same tissues and run a pulsed-field TRF or equivalent method. Flow-FISH for blood would be also an option?

ORIGINAL COMMENT:

-Why is p21 suppressed in the mouse in which the catalytically inactive TERT protein is expressed? The explanation provided by the authors is possible but not experimentally supported. Considering that the TERTCI was intended to be a negative control this result remains concerning in the context of this paper. Resolving the issues around the non-catalytic roles of TERT, if they exist at all, is beyond the scope of this paper, yet can the authors provide their interpretation? Why would p21 stay low in the TERTCI cells once telomeres are exhausted?

AUTHOR RESPONSE:

We address this concern in our general comments and in our answers to referee 1. We now show also by immunofluorescence that p21 levels are greatly reduced in p21+/Tert mice and also in p21+/TertCI mice suggesting that a non-canonical function of Tert reduces p21 levels. We are very interested in this result especially since TertCI reduces p21 levels also in other tissues. Beyond the scope of this manuscript, one hypothesis would be that Tert and TertCI may attenuate the DDR by stimulating a phosphatase through a direct interaction. This is the subject of ongoing and future investigations.

NEW COMMENT by reviewer 2:

It seems possible that the reason this phenomenon is related to the transgene insertion into the p21 locus. The mCherry experiment suggested by reviewer 3 could address this. I understand that experiment will take a long time. Thus, it might need to be later. Yet, I see some risk that the observed effect is not related to telomere length or telomere dysfunction.

ORIGINAL COMMENT:

- Please provide supporting citation for: "C57BL/6NR mice used in this study naturally develop age-related emphysema and mild fibrosis."

AUTHOR RESPONSE:

Ongoing studies in the lab show that a combination lung emphysema (Mean linear intercept) and mild lung fibrosis (Ashcroft score) develop in aged mice. Please find enclosed these results which are provided for the reviewer only.

NEW COMMENT by reviewer 2:

Hmm, it had originally sounded like the authors just forgot to add a citation. This data should be included in the manuscript or be available as a peer reviewed data by the time this work is published.

ORIGINAL COMMENT:

-The premise for the CD34 experiments is quite difficult to follow.

The authors imply that the role of CD34s is the same in acute tissue repair and in telomere length related ageing. Can the authors provide evidence for this? How about the inverse question?

AUTHOR RESPONSE:

We do not know whether Cd34+ cells play the same role in lung repair after injury as they would do in attenuating aging related to telomere shortening? We hypothesize that Tert expression combined with reduced p21 expression maintains a population of Cd34+ cells with proliferative capacity. The exact nature of these cells remains to be determined.

ORIGINAL COMMENT:

How does bleomycin impact CD34s in the p21 TERT model?

AUTHOR RESPONSE:

See our answer to referee 1

ORIGINAL COMMENT:

The telomere length analysis in Figure 1 were performed on lung parenchyma raising the question about the telomere length status of specifically the CD34 population? Can these cells be sorted and analyzed by telomere FISH to elucidate why this population is spared?

AUTHOR RESPONSE:

This is an excellent suggestion, but we do not have any old mice in production to perform this experiment. We will consider this in the future when a new cohort of old mice is available.

NEW COMMENT by reviewer 2:

That is too bad, as this analysis could have addressed many of my concerns outlined above.

Minor points:

ORIGINAL COMMENT:

The Ashcroft score ranks from 0 to 8. A description of what the difference between 2 and 3 on this scale corresponds to would be helpful to the reader.

AUTHOR RESPONSE:

We referred to the original article.

NEW COMMENT by reviewer 2:

So this means that the mice are comparatively, ok? Why is the Kaplan-Meier analysis not included in the ms? This seems very relevant information to benchmark the size of the observed effect of the age related emphysema in B6?

ORIGINAL COMMENT:

Why does the TERTCI show a significant difference in 2D? How can there be no difference in 2D but in 2F? Please explain.

AUTHOR RESPONSE:

Figure 2D is the quantification of fibrosis in the lung parenchyma while Figure 2F reflects peribronchial and perivascular fibrosis. Only the latter two are reduced in p21+/Tert mice.

ORIGINAL COMMENT:

Please state if the histology scoring was performed blinded and by a pathologist?

AUTHOR RESPONSE:

Yes, the experiments were made blind by Larissa Lipskaia who has the relevant expertise.

ORIGINAL COMMENT:

In figure 1B what is the difference between lanes 1 through 11 and why is there such a difference in the lanes? I am sorry, but I do not understand how this experiment was done.

AUTHOR RESPONSE:

Lanes 1 to 11 represent mouse ES mTert^{-/-} clones transfected with a plasmid expressing the mCherry-2A-mTert cassette under the control of the constitutive CAG promoter. Overall, all transfected ES mTert^{-/-} clones have more activity than non-transfected control cells even though the telomerase activity varies from one transfected clone to another.

NEW COMMENT by reviewer 2:

"The lung is one of the few organs where clinical phenotypes exist in humans when telomeres are shortened beyond a critical point." I am not sure that this statement is correct, as we know of many tissues that show pathologies in patients with DC, coats plus and HH.

Referee #3:

The manuscript has been improved and I believe that it is now suitable for publication although a number of points remain unclear to me and this will spark more studies by this group and others.

Minor comments:

- Claiming that "More than 20 000 telomeres distributed in the lung parenchyma were analyzed for each of the 3 mice of the 4 genotypes" sounds grand but it means that, since a cells has ~100 telomeres and 12 mice were studied, only ~ 16 cells per mouse were studied ($20.000/100/3/4 = 16.6$) which is acceptable but at the low end of robustness.
- Cd34 and CD34 are interchangeably used throughout the text.

Referee #1:

This referee would like to commend the authors on the substantial work done to address most of the minor and major concerns raised from the original manuscript. Especially the addition of the hypoxia model of stress-induced emphysema (Figure 9/10) demonstrating the rescued phenotype of p21⁺/Tert mice, validates further their main observations.

We thank referee 1 for this comment.

However, there are still outstanding points that have not been addressed sufficiently. More importantly, however, some of the new data raises further concerns that their model is not functioning as hypothesized. It is already known that telomere dysfunction plays a crucial role in lung aging and COPD and the authors acknowledge it by citing recent reviews and research work (Birch et al 2015, Rossiello et al. 2022), but their new TIF/p21 data suggest otherwise. They "assume that induction of the mTert expression from activated p21 promoter might repair telomeres" but their data do not support this hypothesis. This raises potential issues or technical limitations of their model. More specifically, the study reveals that p21 promoter driven TERT protects against development of ultrashort telomeres, but not against development of dysfunctional telomeres (or from p21 upregulation). However, ultrashort telomeres do not activate cellular senescence, dysfunctional telomeres do. Therefore, unless the authors have evidence that ultrashort telomeres cause cellular senescence or cellular dysfunction in the absence of telomere dysfunction, the beneficial effects of TERT expressed from the p21 promoter (ie. reduced senescence, a decline in capillary density and protection from lung emphysema development) must be due to a function of TERT that does not involve its ability to suppress cellular senescence due to telomere dysfunction. Their conclusion that "TERT delays senescence ... by counteracting telomere shortening..." therefore is likely incorrect.

Furthermore, given the new data demonstrating that p21⁺/TERT mice develop fewer p16 and SA-bGal positive cells, it seems that p21⁺/TERT suppresses cellular senescence in a manner that is dependent on functional telomerase activity but independent of its ability to protect cells from developing dysfunctional telomeres.

These data are difficult reconcile with our current understanding of telomerase function.

We embrace the concerns of this reviewer. Indeed, it was difficult to reconcile the protective role of TERT against ultra-short but not dysfunctional telomeres. We put a lot of effort in resolving this issue. Firstly, we would like to point out that ultra-short telomeres are considered dysfunctional because they do not bind sufficient amount of shelterin and thus are unable to block DDR at telomeres. So why could not we see that TERT protects against dysfunctional telomeres? The problem could be two-fold: (1) the PNA probe used in FISH to detect telomeres is known to have a sensitivity limit which prevents detection of ultra-short telomeres that we reveal by TeSLA; (2) the 53BP1 while being a good marker of DNA double-strand breaks could not be ideal for the analysis of telomere damage in the lungs that most likely arise due to

oxidative stress. There is not much that we can do with the former but we can deal with the latter issue.

Thus, we now present new results of the analysis of the oxidative DNA damage in the lungs, which clearly shows the protective role of TERT (but not TERT^{Cl}). The details are outlined below. That being said, we see no mystery in the fact that TERT reduces cellular senescence in the lungs. We simply had a technical problem with detection of dysfunctional telomeres. We hope the reviewer appreciates our efforts in resolving this issue.

- As demanded by the editor and by the referees, to “*make it clear that this is a lung p21 (and Cd34) paper and not a telomere length paper*”, we have rewritten the abstract as follows:

Lung diseases develop when telomeres shorten beyond a critical point. We constructed a mouse model in which the catalytic subunit of telomerase (mTert), or its catalytically inactive form (mTert^{Cl}), is expressed from the p21^{Cdkn1a} locus. Expression of either TERT or TERT^{Cl} reduces the global p21 level in the lungs of aged mice, highlighting TERT non-canonical function. However, only TERT reduces accumulation of very short telomeres, oxidative damage, endothelial cells (ECs) senescence and senile emphysema in aged mice. Single-cell analysis of the lung reveals that p21 (and hence TERT) is expressed mainly in the capillary ECs. We report that a fraction of capillary ECs marked by CD34 and endowed with proliferative capacity declines drastically with age, and this is counteracted by TERT but not TERT^{Cl}. Consistently, only TERT counteracts decline of capillary density. Natural aging effects are confirmed using the experimental model of emphysema induced by VEGFR2 inhibition and chronic hypoxia. We conclude that catalytically active TERT prevents exhaustion of the putative CD34+ EC progenitors with age, thus protecting against capillary vessel loss and pulmonary emphysema.

It is understandable that several controls missing from some panels were due to unprecedented circumstances during the pandemic, but the absence of key controls makes the interpretation of the in vivo data partially inconclusive.

See our point-by-point response below.

Major outstanding concerns summarized below:

1. The relationship between levels of p21 expression, levels of the p21-mCherry-Tert transgene and telomerase activity remain still unexplored.

We analyzed expression of the Tert KI allele and total Tert by RT-qPCR separately. We came to the conclusion that most of Tert expression is driven by the KI allele. The apparent discrepancy between the high level of p21-mCherry-Tert allele upregulation and only modest increase in telomerase activity could be explained by limited availability of the factors involved in telomerase biogenesis and intracellular trafficking.

In Figure 1D levels of mCherry-Tert are shown to be ~10 fold higher in 3-month old p21^{+/Tert} mice compared to p21^{+/+} mice the p21^{+/+} although their respective Cdkn1a levels are shown to be similar. This means that the mTert might not be entirely induced in a p21-dependent manner. This is a main point not addressed in the rebuttal.

As mentioned above, the new experiments in the manuscript suggest a feedback mechanism (the mode of action of which remains to be discovered) whereby Tert independently reduces p21 gene expression and p21 protein levels. This likely explains the results of Fig. 1D.

2. As mentioned in the previous review, the p21/Tert construct used in the mouse model needs to be evaluated more carefully in cell cultures and/or in the mouse. The pCAG-controlled mCherry-mTert construct used for Fig 1B is not appropriate for evaluating enzymatic activity as it differs from the construct used in the mouse.

We have evaluated DDR-dependent *mCherry-2A-Tert* induction both *in vitro* and *in vivo* (all these data are in the manuscript). Concerning Fig 1B, our aim was to evaluate the catalytic activity of the TERT encoded by the cassette used to generate p21^{+/Tert} mouse.

The key control is whether the p21-induced Tert transgene in the mouse develops enzymatic activity upon p21 promoter activation. These data, using PA-SMC cells, should be shown in the main Figure.

We have kept the Figure 1 and EV2 as they are but we have emphasized in the text that telomerase catalytic activity is increased upon p21 promoter activation. At the same time, we answer in the text the point below to explain the apparent discrepancy between the level of induction of the transgene and the 2-fold increase in telomerase activity. (See next point).

We write in the first paragraph:

Importantly, we also detected a two-fold higher telomerase activity in p21^{+/Tert} cells at passages 2-4, before PA-SMCs cumulative PDL curve became significantly different (Fig. EV2C) indicating that upregulation of mTERT alone is sufficient to increase telomerase activity. We explain the apparent discrepancy between the high level of transgene induction and only 2-fold increase in telomerase activity by the fact that telomerase biogenesis and trafficking require numerous factors that can be limiting.

Proper controls are cells from all genotypes including from cells expressing the wt and mutant Tert. P21 promoter should be activated due to replicative aging as well as due to DNA damaging drugs. Tert activity and mRNA expression levels should be measured in all cases, especially in

light of the observations that Tert activity and mRNA expression seem to be uncoupled (see Figure EV2 in Panels A-C where the authors show ~80-fold induction of the transgene mRNA in the cultured PA-SMCs, but only a 2-fold increase in telomerase activity. Why is this?).

It is thought that in mouse the level of catalytic subunit of telomerase rather than its RNA component limits the activity. However, telomerase biogenesis and trafficking (spatially regulated assembly of the holoenzyme) is extremely complex and other accessory factors (dyskerin, NOP10, NHP2, and GAR1) as well as additional cofactors required for the assembly of telomerase RNP may limit the activity in vivo. Therefore, we are not surprised that a large increase in the Tert mRNA leads to only two-fold increase in activity.

These are not trivial questions, since all the conclusions of the in vivo data regarding the mechanism of how p21 induced mTert alleviates emphysema in the old lung are based on the premise that the transgene is expressed and translated properly, that the transgene has enzymatic activity, and that enzymatic activity is increased upon DNA damage as well as in response to telomere dysfunction.

We hope that the reviewer is satisfied and convinced now that “the transgene is expressed and translated properly, and provides enzymatic activity upon p21 activation. We invite referee 1 to consider our comments outlined above.

3. The Kaplan-Meier survival curves included in rebuttal show no statistical differences between groups, but do show clear trends of increased mortality in the mice p21+/-, p21+/Tert (50% survival rate). This needs to be discussed in the text, since Tert expressed from a p21 promoter (or p21 haploinsufficiency) does not appear to benefit the organism as a whole.

We have a comprehensive study showing that with age, a proportion of p21+/Tert mice develop hepatocarcinoma and to less extent lymphoma. This could explain the Kaplan-Meier survival curves. This is why we prefer not to show Kaplan-Meier survival curves in this study.

4. If p21 protein levels are perturbed in all genotypes, and more so in both mTert, mTertCI, as shown in the new Figure 2A it demonstrates that p21 mediated senescence is perturbed (Figure 5E,F SAbgal trends lower in p21+/TertCI compared to p21+/-). If “the p21+/TertCI mice exhibit the same aging characteristics in the lung as control mice” it suggests that DNA damage-induced p21-mediated senescence is not a main contributor in age-related lung emphysema, which contradicts the main idea revolved around the generation of the transgene.

See our answer at the end of page 2: “In summary, because p21 is required for the”

The authors should analyze abundance of total DDR foci, regardless whether they are telomeric or not, as the data can be easily extracted from their TAF/TIF analysis. A decrease of DDR foci in p21/TERT animals would suggest that TERT exerts its beneficial effects due to its non canonical function in repairing non-telomeric DNA damage.

We quantified the total 53BP1 foci, as requested, but still couldn't find the effect of Tert expression. For the reasons explained above and in the revised manuscript, we have analyzed instead the global level of oxidative DNA damage by measuring 8-oxo-dG levels using

validated anti-8-oxo-dG antibody (New Fig. 2C). We found that the global level of oxidative damage was reduced in the lungs of p21^{+Tert}, but not p21^{+TertCI} mice (Fig. 2C). We believe that TERT protects the genome from global oxidative damage by reducing ROS activation by mitochondria which is triggered by telomere oxidative damage, a vicious cycle described by Passos and Zglinicki (references are included in the revised version).

5. In EV3 graph comparisons of VSTs of p21^{+/+} vs p21^{+/-} and p21^{+TertCI} must also be shown to understand at what levels haploinsufficiency alone affects the distribution of VSTs (the data are there).

Currently, we can only compare (p21^{+/+} vs p21^{+Tert}) and (p21^{+/-} vs p21^{+TertCI}).

6. The authors should speculate less about the non-telomeric roles of mTert in regulating p21 and CD34, since it's also possible that the mTERT-p21 mRNA hybrid produced from the cassette has additional functions that are confounding to the function of an inducible native mTert - even a possible dominant negative form that affects normal levels of p21.

This is done in the discussion, see above.

Although, the addition of the hypoxia mouse model in the revised manuscript provides solid evidence that indeed mTERT can rescue induced emphysema (the main scope of the study), their aging model presents several major limitations (as discussed above) that need to be addressed.

We hope the additional revision will convince referee 1.

Referee #2:

Comments to revision by reviewer 2:
The authors provided more explanations and arguments without yet did not provide the additional experiments that I asked for. Below I try to clarify my points and requests.

In summary, I am still concerned that the mild effect may be pleiotropic, as clear evidence for telomere length or telomere dysfunction drives the effects is missing.

See my comments to referee 1.

In addition, the effects on the p21 levels and on the persistence of CD34⁺ positive cells are not mild at all. These are very clear effects.

That said, I appreciate the significant amount of work that went into this manuscript. I agree with the authors that the chronic hypoxia data supports their observations, yet again the experiment does not provide insight into the mechanism(s) at play.

Thank you for your comments.

ORIGINAL COMMENT:

-The telomere length analysis is difficult to interpret. It is of course correct that the short 11 telomeres are most relevant but the assay used is strongly biased towards short telomeres. The TESLA assay in the experiment analyzes the telomeres of ~80pg of genomic DNA = ~12 cells per reaction with ~100 telomeres/ cell. Each lane shows about >12 telomeres. This suggests that ~1% of the telomeres in these mice are short. This is unexpected.

AUTHOR RESPONSE:

In our hands, TeSLA amplifies telomeres up to ~15 kb. Assuming a bell-shaped distribution (could be skewed) of mouse telomere lengths with an average around 45-50 kb, only a small proportion of telomeres (~5-10% in one tail of the distribution) can be analyzed by TeSLA. Therefore, if 1% of the telomeres amenable to TeSLA analysis are short, it corresponds to only 0.05-0.1% of total mouse telomeres. The validity of TeSLA for the analysis of short telomeres in mice was addressed in the original publication (Lai et al., Nat Comm 2017).

NEW COMMENT by reviewer 2:

I do not agree with the authors on this point. The authors add approximately 1200- 2400 (after S-phase) telomeres in a tube and then detect 12-20 short telomeres? Where do all these telomeres come from? If the authors DNA quantification is equal and accurate, their data clearly puts the Lai et al. publication into question. Stats and analysis need to be provided, as well as the base assumptions about the telomere length (distribution) above 15kb and how this was included in the stats. These experiments need independent corroboration.

In our TeSLA workflow, after the 2nd ligation of the subtelomeric adaptors, the DNA is diluted to 500 pg per μL and 1 μL goes to each PCR reaction. Therefore, PCR is initiated with $500/2.9=172.4$ haploid genome equivalents or $172.4 \times 40 = 6,896.6$ telomeres. There is absolutely nothing surprising that we detect 12-20 telomeres shorter than 15 kb.

ORIGINAL COMMENT:

The authors need to evaluate this using more standard techniques that have less of a telomere length bias (FISH, pulsed field TRF, etc.).

Yes, there is a small yet appreciable difference in the telomere length in the TERT-p21 het cells. It seems important that the authors establish the constitutional telomere length is the same in all mouse strains. This is to control for the additional copy of TERT elongates -driven by p21- did not increase telomere length in the germline and to control for founder effects during the generation of the mice. In other words, do the authors have any evidence that the p21 regulation matters for the effects seen? Or is the basal p21 expression (-driving TERT) sufficient to elongate telomeres?

AUTHOR RESPONSE:

If we understand well, the reviewer is concerned that telomeres in p21+/Tert mice could be preelongated during either gameto- or embryogenesis due to basal p21 promoter activity. We are not sure that this concern is really relevant, because pre-elongation of telomeres does not protect them from breakage that generates VSTs.

NEW COMMENT by reviewer 2:

I am not sure why the authors say that 'pre-elongation of telomeres does not protect them from breakage that generates VSTs'. Have the authors shown that leads to "the short telomeres" in their model? And even if it was the mechanism, longer telomeres will break into larger residual telomeres don't they? I am not sure why the authors argue against a trivial experiment that takes 3-4 days to perform to check for this. My request is also motivated by the apparent artifacts reveal by the authors TESLA analysis. Thus, please provide telomere length data using non-TESLA data for somatic tissue ear, tail and lung or anything else that demonstrated that "starting telomere length (see explanation above) that telomere length in these mice in not altered in young mice across genotypes. I suggest to take DNA from 4 mice (young age matched) of each genotype (4) harvest DNA of the same tissues and run a puls-field TRF or equivalent method. Flow-fish for blood would be also an option?

We performed a pulse-field TRF Southern blot to assess bulk telomere length in mice of the 4 genotypes. We didn't find a difference (at the level of TRF Southern blot resolution) in the mean telomere length between the 4 genotypes. Therefore, we can conclude that no substantial changes to mean telomere length occurred in the germline or during embryonic development of the model mice. We enclose this figure as a new dataset 2 Fig. 1 (page 4, end of the first paragraph).

ORIGINAL COMMENT:

-Why is p21 suppressed in the mouse in which the catalytically inactive TERT protein is expressed? The explanation provided by the authors is possible but not experimentally supported. Considering that the TERTCI was intended to be a negative control this result remains concerning in the context of this paper. Resolving the issues around the non-catalytic

roles of TERT, if they exist at all, is beyond the scope of this paper, yet can the authors provide their interpretation? Why would p21 stay low in the TERT^{CI} cells once telomeres are exhausted?

AUTHOR RESPONSE:

Our interpretation is that both TERT and TERT^{CI} reduce the level of p21 via non-canonical function. We propose explanations in the discussion related to the ability of TERT to prevent MYC degradation (Koh et al., JCI 2015), which is a known repressor of the *Cdkn1a* gene. This, in turn, facilitates entry of the CD34⁺ EC progenitors into the cell cycle. However, telomere maintenance becomes crucial during expansion of the CD34⁺ progenitors. This is why no beneficial effect is seen in the lungs of the p21^{=/Tert^{CI}} mice beyond the reduction of p21 level. This is discussed in details in the revised version (references are given therein).

NEW COMMENT by reviewer 2:

It seems possible that the reason this phenomenon is related to the transgene insertion into the p21 locus. The mCherry experiment suggested by reviewer 3 could address this. I understand that experiment will take a long time. Thus, it might need to be later. Yet, I see some risk that the observed effect is not related to telomere length of telomere dysfunction.

As indicated in our response to referee 1, the reduction in p21 levels is not related to telomeric damage, or to general DNA damage, since this reduction is also observed p21^{+/Tert^{CI}} lungs while the level of DNA damage remains high in these mice. Although we don't know the mechanism, we have observed this phenomenon in many other situations. For instance, very recent results from our team show that HFD-fed p21^{+/Tert} and p21^{+/Tert^{CI}} mice have reduced expression of mCherry analyzed by FACS (mCherry is used as a read-out of p21 expression) in the stromal fraction of the adipose tissue compared to p21^{+/mcherry} mice, while gamma-H2AX levels are not decreased in the HFD fed p21^{+/Tert} and p21^{+/Tert^{CI}} mice

ORIGINAL COMMENT:

- Please provide supporting citation for: "C57BL/6NR mice used in this study naturally develop age-related emphysema and mild fibrosis."

AUTHOR RESPONSE:

Ongoing studies in the lab show that a combination lung emphysema (Mean linear intercept) and mild lung fibrosis (Aschcroft score) develop in aged mice. Please find enclosed these results which are provided for the reviewer only.

NEW COMMENT by reviewer 2:

Hmm, it had originally sounded like the authors just forgot to add a citation. This data should

be included in the manuscript or be available as a peer reviewed data by the time this work is published.

We provide in the previous rebuttal results that are not yet published showing that mice develop age-related emphysema and mild fibrosis.

We have added the following reference to answer the concern of referee 2:

Phospholipase A2 receptor 1 promotes lung cell senescence and emphysema in obstructive lung disease. Beaulieu D, Attwe A, Breau M, Lipskaia L, Marcos E, Born E, Huang J, Abid S, Derumeaux G, Houssaini A, Maitre B, Lefevre M, Vienney N, Bertolino P, Jaber S, Nouredine H, Goehrig D, Vindrieux D, Bernard D, Adnot S. Eur Respir J. 2021 Aug 12;58(2):2000752. doi: 10.1183/13993003.00752-2020

ORIGINAL COMMENT:

-The premise for the CD34 experiments is quite difficult to follow. The authors imply that the role of CD34s is the same in acute tissue repair and in telomere length related ageing. Can the authors provide evidence for this? How about the inverse question?

AUTHOR RESPONSE:

We turned our attention to the CD34+ cells because another single cell study (Niethammer et al., eLife 2020) reported that a fraction of capillary ECs expressing *Cd34* was poised to lung regeneration in response to injury. We found that both during natural ageing and in the experimentally induced emphysema model, *Tert* expression preserves the population of CD34+ cells, and this correlates with the maintenance of the capillary density. The exact nature of these cells remains to be determined. I guess the referee is satisfied by this response.

ORIGINAL COMMENT:

How does bleomycin impact CD34s in the p21 TERT model?

AUTHOR RESPONSE:

See our answer to referee 1. I guess the referee is satisfied by this response.

ORIGINAL COMMENT:

The telomere length analysis in Figure 1 were performed on lung parenchyma raising the question about the telomere length status of specifically the CD34 population? Can these cells be sorted and analyzed by telomere FISH to elucidate why this population is spared?

AUTHOR RESPONSE:

This is an excellent suggestion, but we do not have any old mice in production to perform this experiment. We will consider this in the future when a new cohort of old mice is available.

NEW COMMENT by reviewer 2:

That is too bad, as this analysis could have addressed many of my concerns outlined above. I agree, but the situation has been very difficult for us. Please consider that we have demonstrated that:

- 1) CD34+ cell populations are preserved in aged p21^{+/Tert} mice,
- 2) CD34+ cells are also stained with proliferation markers (PCNA, BrdU) but not with senescence markers (p16).
- 3) We have demonstrated the relationship between the presence of CD34+ cells with proliferative activity and protection against emphysema in aged mice.

Minor points:

ORIGINAL COMMENT:

The Ashcroft score ranks from 0 to 8. A description of what the difference between 2 and 3 on this scale corresponds to would be helpful to the reader.

AUTHOR RESPONSE:

We referred to the original article cited in the manuscript.

NEW COMMENT by reviewer 2:

So this means that the mice are comparatively, ok? Yes

Score 2 and 3 are defined as follows:

2)

Alveolar septa: Clearly fibrotic changes (septum >3× thicker than normal) with knot-like formation but not connected to each other

Lung structure: Alveoli partly enlarged and rarefied, but no fibrotic masses

3)

Alveolar septa: Contiguous fibrotic walls (septum >3× thicker than normal) predominantly in whole microscopic field

Lung structure: Alveoli partly enlarged and rarefied, but no fibrotic masses

This is now specified in the MS.

Why is the Kaplan-Meier analysis not included in the ms? This seems very relevant information to bench mark the size of the observed effect of the age related emphysema in B6?

See our answer to referee 1.

ORIGINAL COMMENT: Fig. 2 is now Fig. 3

Why does the TERTCI show a significant difference in 2D? In Fig 2D, there is a tendency but it is not significant. How can there be no difference in 2D but in 2F?

Please explain.

AUTHOR RESPONSE:

Figure 2D is the quantification of fibrosis in the lung parenchyma while Figure 2F reflects peribronchial and perivascular fibrosis. Only the latter two are reduced in p21+/Tert mice. We hope the answer is clear

ORIGINAL COMMENT:

Please state if the histology scoring was performed blinded and by a pathologist?

AUTHOR RESPONSE:

Yes, the experiments were made blind by Larissa Lipskaia who has the relevant expertise.

ORIGINAL COMMENT:

In figure 1B what is the difference between lanes 1 through 11 and why is there such a difference in the lanes? I am sorry, but I do not understand how this experiment was done.

AUTHOR RESPONSE:

Lanes 1 to 11 represent mouse ES mTert^{-/-} clones transfected with a plasmid expressing the mCherry-2A-mTert cassette under the control of the constitutive CAG promoter. Overall, all transfected ES mTert^{-/-} clones have more activity than non-transfected control cells even though the telomerase activity varies from one transfected clone to another. The latter is likely due to the plasmid copy number maintained by the different ES mTert^{-/-} clones.

NEW COMMENT by reviewer 2:

"The lung is one of the **few** organs where clinical phenotypes exist in humans when telomeres are shortened beyond a critical point." I am not sure that this statement is correct, as we know of many tissues that show pathologies in patients with DC, coats plus and HH.

We have deleted "few" to make this sentence correct.

Referee #3:

The manuscript has been improved and I believe that it is now suitable for publication although a number of points remain unclear to me and this will spark more studies by this group and others.

We thank referee 3 for his/her comment. See our comments to referee 1

Minor comments:

- Claiming that "More than 20 000 telomeres distributed in the lung parenchyma were analyzed for each of the 3 mice of the 4 genotypes" sounds grand but it means that, since a cells has ~100 telomeres and 12 mice were studied, only ~ 16 cells per mouse were studied ($20.000/100/3/4 = 16.6$) which is acceptable but at the low end of robustness.

We have analyzed 20 000 telomeres for each mice. According to our calculation (if we consider that each cell contains 92 telomeres), we have analyzed 200 cells/mouse (and not 16,6).

- Cd34 and CD34 are interchangeably used throughout the text. We have standardized the nomenclature by using CD34 to refer to the protein and *Cd34* to refer to the gene or mRNA expression.

Dear Dr. Geli,

Thank you for the submission of your further revised manuscript to our editorial offices. I have now received the reports from two referees that I asked to re-evaluate your study, you will find below. As you will see, both referees support the publication of the study in EMBO reports. Referee #1 has remaining points, I ask you to address in a final revised manuscript. Please also provide a final p-b-p-response regarding these points.

Moreover, I have these editorial requests I ask you to address:

- Please reduce the number of keywords to 5.
- We updated our journal's competing interests policy in January 2022 and request authors to consider both actual and perceived competing interests. Please review the policy <https://www.embopress.org/competing-interests> and update your competing interests if necessary. Please name this section 'Disclosure and Competing Interests Statement' and put it after the Acknowledgements section.
- Please add scale bars of similar style and thickness to the microscopic images (main, EV and Appendix figures), using clearly visible black or white bars (depending on the background). Please place these in the lower right corner of the images themselves. Please do not write on or near the bars in the image but define the size in the respective figure legend. There are still figures without scale bars, or hardly visible scale bars, or scale bars with text nearby.
- Could statistical testing be provided for the diagrams shown in Figs. 1D, 3D, 7F, EV1D and EV2A/B?
- Please provide a proper Appendix file. The Appendix should have page numbers and needs to include a table of content on the first page (with page numbers) and legends for all content. Please follow the nomenclature Appendix Figure Sx, Appendix Table Sx etc. throughout the text (each item needs to be called out), and also label the figures and tables according to this nomenclature. Presently, the Appendix contains several images without defining them as individual figures. These need to be grouped into figures with title, legend and callout.
- Please name the dataset uploaded as 'Data Set Figure 4' Dataset EV1. Please change its legend accordingly and add a callout to the manuscript text file.
- What is 'Data Set 2 Figure 2'? Is this source data or a table that is called out in the text? If this is source data, please add this to the source data for Fig. 2. Otherwise, I would suggest to move this as Appendix table to the Appendix (Appendix Table S1) and to call it out in the manuscript text accordingly.
- There are two files uploaded named 'Dataset 1 Figure 1' and 'Dataset 2 Figure 1'. These do not contain information in dataset format, but rather figures. I would suggest to move these into the Appendix. Please arrange these as Appendix figures with title, legend and add callouts to the manuscript text.
- Please make sure that all the funding information is also entered into the online submission system and that it is complete and similar to the one in the acknowledgement section of the manuscript text file. It seem that information regarding the European Union's Horizon 2020 research and innovation programme under the Marie Skłodowska-Curie grant agreement No 666003; Ministère de la Recherche; EDF (CT9818) and ITMO Cancer is still missing from the submission system.
- Please note our reference format (we need up to 10 authors listed before 'et al'). See: <http://www.embopress.org/page/journal/14693178/authorguide#referencesformat>

Referee #1:

In this revised manuscript, the authors have now included additional data that support a role of telomerase in protecting from age associated-capillary vessel loss and pulmonary emphysema through both its canonical and non-canonical functions. Conclusions were adjusted appropriately. While no more major issues persist, the authors should address the following minor issues:

- 1) Figure 2B lacks description of y-axis. Figure legend 2B is shown twice. Also, the legend states that images were acquired in z-sections spaced 10 microns apart using a 40x oil lens. However, in the methods section the authors state that the entire tissue section is only 5 microns thick and that images were acquired using a 60x lens. Please correct these discrepancies. Also, if z-sections were indeed imaged 10 microns apart, it is no surprise that only few TIFs were detected as only very narrow regions in imaged cell nuclei were analyzed.
- 2) Figure 5B and D show quantitation of p16 expressing cells. Judging from the representative images shown, it seems that p16 signal is somewhat ambiguous and difficult to quantify. Yet, data points cluster tightly in the bar graphs. The authors should report how these data were acquired. For example, were images evaluated in a blinded manner by multiple individuals? If not, how was accuracy confirmed and what were the criteria to designate a cell p16+ vs p16-?
- 3) Figure 4E: figure legend describes colors that are not accurate. Please revise.
- 4) Figure 5 scale bars are either missing, or not described in the legend. This is also observed in other figures illustrating IF or IHC tissue sections. Please correct throughout the manuscript.
- 5) Figure 9: please describe in legend or methods how MLI was evaluated.

Referee #2:

This paper has been revised several times from its initial submission in EMBO J. This paper is suitable in its current form but it is important that the full reviews starting at the EMBO J submission are made available to contextualize this paper.

Referee #1:

In this revised manuscript, the authors have now included additional data that support a role of telomerase in protecting from age associated-capillary vessel loss and pulmonary emphysema through both its canonical and non-canonical functions. Conclusions were adjusted appropriately. While no more major issues persist, the authors should address the following minor issues:

1) Figure 2B lacks description of y-axis. **Corrected.**

Figure legend 2B is shown twice. **Corrected.**

Also, the legend states that images were acquired in z-sections spaced 10 microns apart using a 40x oil lens. However, in the methods section the authors state that the entire tissue section is only 5 microns thick and that images were acquired using a 60x lens. Please correct these discrepancies. Also, if z-sections were indeed imaged 10 microns apart, it is no surprise that only few TIFs were detected as only very narrow regions in imaged cell nuclei were analyzed.

The tissue section is indeed 5 microns thick. 10 sections of 1 μ m have been acquired and section with signal have been projected (maximum intensity). The legend has been modified as follows: *Images represent the Maximum intensity projection of the 5 μ m section taken with a \times 60 oil objective.*

2) Figure 5B and D show quantitation of p16 expressing cells. Judging from the representative images shown, it seems that p16 signal is somewhat ambiguous and difficult to quantify. Yet, data points cluster tightly in the bar graphs. The authors should report how these data were acquired. For example, were images evaluated in a blinded manner by multiple individuals? If not, how was accuracy confirmed and what were the criteria to designate a cell p16+ vs p16-?

The images were evaluated in a blinded manner by 2 different persons. Cells with a white focus that stands out clearly from the background are considered p21-positive.

3) Figure 4E: figure legend describes colors that are not accurate. **Corrected**

4) Figure 5 scale bars are either missing, or not described in the legend. This is also observed in other figures illustrating IF or IHC tissue sections. Please correct throughout the manuscript. **Done, see above.**

5) Figure 9: please describe in legend or methods how MLI was evaluated. **In Materials and Methods we mention a reference for measuring MLI (Crowley *et al*, 2019).**

- All editorial and formatting issues were resolved by the authors.

Dr. Vincent Geli
Centre National de la Recherche Scientifique
Center of Research in Cancerology of Marseille (CRCM)
27 bd Leï Roure
Marseille, Cedex 09, BP 30059, BDR 13273
France

Dear Dr. Geli,

I am very pleased to accept your manuscript for publication in the next available issue of EMBO reports. Thank you for your contribution to our journal.

Yours sincerely,
